# Cloud droplet formation at the base of tropical convective clouds: closure between modeling and measurement results of ACRIDICON–CHUVA

Ramon Campos Braga[1], Barbara  Ervens[2], Daniel Rosenfeld[3], Meinrat O. Andreae[4,5], Jan-David Förster[1], Daniel Fütterer[6], Lianet Hernández Pardo[1], Bruna A. Holanda[1], Tina Jurkat - Witschas[6], Ovid O. Krüger[1], Oliver Lauer[1], Luiz A. T. Machado[1,7], Christopher Pöhlker[1], Daniel Sauer[6], Christiane Voigt[6,8], Adrian Walser[6], Manfred Wendisch[9], Ulrich Pöschl[1], and Mira L. Pöhlker[1]

[1]Multiphase Chemistry Department, Max Planck Institute for Chemistry, 55128 Mainz, Germany
[2]Université Clermont Auvergne, CNRS, SIGMA Clermont, Institut de Chimie de Clermont-Ferrand, 63000 Clermont-Ferrand, France
[3]Institute of Earth Sciences, The Hebrew University of Jerusalem, 9190401 Jerusalem, Israel.
[4]Biogeochemistry Department, Max Planck Institute for Chemistry, 55128 Mainz, Germany
[5]Scripps Institution of Oceanography, University of California San Diego, La Jolla, CA 92037, USA
[6]Institute of Atmospheric Physics, German Aerospace Center (DLR), 82234 Oberpfaffenhofen, Germany
[7]National Institute for Space Research (INPE), 12227-010 São José Dos Campos, Brazil
[8]Johannes Gutenberg University Mainz, 55099 Mainz, Germany
[9]Faculty of Physics and Earth Sciences, Leipzig Institute for Meteorology, University of Leipzig, 04103 Leipzig, Germany

**Correspondence:** Mira L. Pöhlker (m.pohlker@mpic.de) and Barbara Ervens (barbara.ervens@uca.fr)

**Abstract.**

Aerosol-cloud interactions contribute to the large uncertainties in current estimates of climate forcing. We investigated the effect of aerosol particles on cloud droplet formation by model calculations and aircraft measurements over the Amazon and over the western tropical Atlantic during the ACRIDICON-CHUVA campaign in September 2014. On the HALO research

aircraft, cloud droplet number concentrations ($N_d$) were measured near the base of clean and polluted growing convective cumuli using a cloud combination probe (CCP) and a cloud and aerosol spectrometer (CAS-DPOL). An adiabatic parcel model was used to perform cloud droplet number closure studies for flights in differently polluted air masses. Model input parameters included aerosol size distributions measured with an ultra-high sensitive aerosol spectrometer (UHSAS), in combination with a condensation particle counter (CPC). Updraft velocities ($w$) were measured with a boom-mounted Rosemount probe. Over

the continent, the aerosol size distributions were dominated by accumulation mode particles, and good agreement between measured and modeled $N_d$ values was obtained (deviations $\lesssim$10%) assuming an average hygroscopicity of $\kappa \sim 0.1$, which is consistent with Amazonian biomass burning and secondary organic aerosol. Above the ocean, fair agreement was obtained assuming an average hygroscopicity of $\kappa \sim 0.2$ (deviations $\lesssim 16\%$) and further improvement was achieved assuming different hygroscopicities for Aitken and accumulation mode particles ($\kappa_{\mathrm{Ait}} = 0.8$, $\kappa_{\mathrm{acc}} = 0.2$; deviations $\lesssim 10\%$), which may reflect

secondary marine sulfate particles. Our results indicate that Aitken mode particles and their hygroscopicity can be important for droplet formation at low pollution levels and high updraft velocities in tropical convective clouds.

# 1 Introduction

Aerosol-cloud-interactions represent one of the largest uncertainties in our current understanding of the Earth's climate system, according to the latest report by the Intergovernmental Panel on Climate Change (IPCC, 2021). Aerosols have a strong effect on cloud properties since cloud droplets form on cloud condensation nuclei (CCN) by condensation of water vapor. The uptake of water vapor by aerosol particles and the subsequent activation and growth of cloud droplets is described by the Köhler theory (Köhler, 1936), relates the water vapor saturation ratio ($s$) to the water activity in the aqueous solution ($a_{\mathrm{w}}$, *Raoult term*), which is the size and composition dependencies of the droplet's solute effect, as well as the increase in equilibrium water vapor pressure due to droplet's surface curvature (*Kelvin term*). The Raoult effect (solute term) is commonly parameterized using the hygroscopicity parameter $\kappa$ with values ranging from $\sim 0.1$ to $\sim 0.9$ for single components of atmospheric aerosol particles (Petters and Kreidenweis, 2007). The $\kappa$ values of ambient aerosol particles are in the range of $\sim 0.09 < \kappa < 0.18$ for the complex mixtures of organic aerosols and $0.1 < \kappa < 0.3$ for biomass burning aerosol (e.g., Andreae and Rosenfeld, 2008; Carrico et al., 2010; Engelhart et al., 2012).

Many CCN closure studies in various parts of the world have been performed to improve our understanding of the relationship between aerosol properties and their ability to form cloud droplets (e.g., Rissler et al., 2004; Broekhuizen et al., 2006; Wang et al., 2008; Ervens et al., 2010). Such closure studies compare the predicted CCN number concentration ($N_{CCN}$) according to Köhler theory based on particle size and composition (hygroscopicity) to results from CCN measurements, i.e. for equilibrium conditions at different supersaturations in CCN counters. Much fewer studies compare predicted ($N_{d,p}$) and measured cloud droplet number concentrations ($N_{d,m}$) ('cloud droplet number closure'), since often direct measurements or estimates of the updraft velocities ($w$) near cloud base are not available. $N_{d,p}$ at cloud bases is commonly calculated in adiabatic cloud models based on the hygroscopic growth of CCN particles with a prescribed $\kappa$ and $w$. These models simulate the expansion and cooling of air, the resulting changes in relative humidity, and the condensational growth of cloud droplets (Reutter et al., 2009; Ervens et al., 2010; Leaitch et al., 2010). Updraft velocity is often used as a fitting parameter to match $N_{d,m}$ (e.g., Anttila et al., 2012). Other studies used $w$ distributions to predict a range of $N_{d,p}$ (Chuang et al., 2000; Peng et al., 2005; Meskhidze et al., 2005; Hsieh et al., 2009). Previous cloud droplet number closure studies suggested that $w$ is one of the most poorly constrained parameters leading to large uncertainties in the predictions of $N_{d,p}$ (e.g., Conant et al., 2004; Fountoukis et al., 2007).

The Amazon Basin is a unique region to test our understanding of aerosol-cloud interactions in shallow and deep convective clouds due to large variability in aerosol concentration during the dry and wet seasons (e.g., Artaxo, 2002; Andreae et al., 2004; Pöhlker et al., 2016, 2018). The properties and dynamics of clouds over this pristine rain forest region can be fundamentally changed by anthropogenic emissions (e.g., Roberts et al., 2003; Rosenfeld et al., 2008; Reutter et al., 2009; Pöhlker et al., 2018). To explore aerosol-cloud interactions, cloud microstructure and precipitation-forming processes above the Amazon rain forest, the ACRIDICON-CHUVA (Aerosol, Cloud, Precipitation, and Radiation Interactions and Dynamics of Convective Cloud Systems - Cloud processes of tHe main precipitation systems in Brazil: A contribUtion to cloud resolVing modeling

and to the GlobAl Precipitation Measurements) campaign with the HALO (High Altitude Long Range Research) aircraft took place during the dry season in September 2014 (Wendisch et al., 2016).

The focus of this study is to describe a closure analysis based on the $N_{d,m}$ at cloud bases of convective clouds and $N_{d,p}$ calculated from an adiabatic parcel model (Feingold and Heymsfield, 1992; Ervens et al., 2005) that is based on other independently measured properties. To this end, measurements of droplet concentrations at cloud bases performed during the ACRIDICON-CHUVA campaign were used. Furthermore, our calculations explicitly simulated the condensational growth of aerosol particles from below cloud base up to a height of several meters above the level, at which they grow to cloud droplet sizes. We compared $N_{d,m}$ at cloud bases of convective clouds in different air masses with $N_{d,p}$, using in situ measurements of aerosol size distributions as model input.

The closure analysis was performed separately for two cloud probes (Cloud droplet number concentrations and size distributions were measured by a Cloud Combination Probe – Cloud Droplet Probe, CCP-CDP, and Cloud and Aerosol Spectrometer with Depolarization, CAS-DPOL) mounted onboard HALO (Wendisch et al., 2016; Voigt et al., 2017). This was performed to verify our methodology using two types of instruments to measure number concentrations of droplets with different particle inlet characteristics and uncertainties. Our study was performed to explore the sensitivities of effective particle hygroscopicity ($\kappa$), aerosol particle number concentration ($N_a$), and $w$ to $N_d$. The results reveal the sensitivity of $N_d$ to $\kappa$, $N_a$ and $w$ for measurements over the Amazon basin during the dry season and over Atlantic Ocean.

## 2 Aircraft Measurements

The measurements were performed aboard the High Altitude LOng Range aircraft (HALO), a modified business jet G550 (manufactured by Gulfstream, Savannah, USA). In situ meteorological and avionics data, such as the vertical velocity, were obtained at 1 Hz from the BAsic HALO Measurement And Sensor System (BAHAMAS). A boom-mounted Rosemount model 858 AJ air velocity probe was used to measure the updraft velocity with BAHAMAS, measuring in a range of 0.1 m s$^{-1}$ $\leq$ $w \leq$ 6 m s$^{-1}$. The uncertainties in measured $w$ are $\Delta w <$ 0.2 m s$^{-1}$ for $w <$ 5 m s$^{-1}$ and $\Delta w \approx$ 0.25 m s$^{-1}$ for $w >$ 5 m s$^{-1}$. Further details on the uncertainties of $w$ measurements are described by Mallaun et al. (2015). The measurements took place over the Amazon Basin and over the western tropical Atlantic in September 2014 during the ACRIDICON–CHUVA campaign (Wendisch et al., 2016).

Figure 1a shows the measurement region for the flights analyzed in this study (the flights are labelled with 'AC' and a running number, in agreement with the naming e.g., (Wendisch et al., 2016)). The region of cloud base measurements is indicated by circles for each flight. The measurement strategy was developed such that measurements were made within at most 10 minutes and 60 km from each other. This was performed to assure that droplet measurements at cloud base pertain to the same air mass as the aerosol measurements below cloud base. A *conceptual* representation of the cloud profiling, including flight legs below and within cloud base, is shown for measurements during flight AC19 in Fig. 1b. For the present study, the flight legs below and at cloud base are of primary relevance. Such flight legs, during which the relevant aerosol and cloud microphysical data were obtained, are distinguished by different colors. During flight AC19 the profiling of the marine shallow cumulus clouds

was conducted up to an altitude of 4.3 km; details on the air mass origin and the aerosol properties during this flight can be found in Section 2.1. Aerosol properties were investigated during the flight leg below cloud base, which had a length of about

19 km at an altitude of $\sim 450$ m above sea level (asl). Cloud microphysical properties of the marine shallow cumulus clouds (Atlantic ocean) were investigated during the flight leg near cloud base, which had a length of $\sim 60$ km at an altitude at $\sim$ 600 m asl. A similar strategy was applied for in-land flights. Convective cumuli formed in very polluted environments (arc of deforestation) directly above the Amazonian deforestation arc during flight AC07. Less polluted clouds were found farther away from the deforestation fires over the tropical rain forest (remote Amazon) during flights AC09 and AC18.

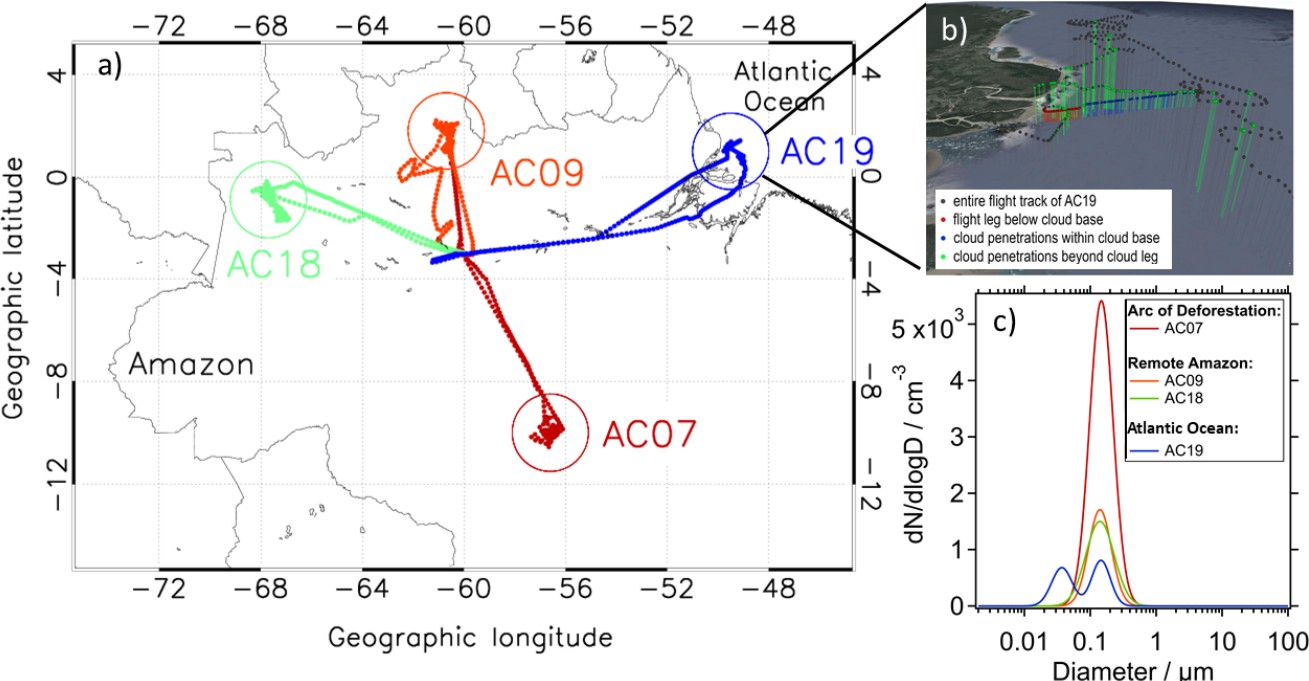

**Figure 1.** a) HALO flight tracks during the ACRIDICON–CHUVA experiment, color-coded for the different flights. Circles indicate the region of aerosol and cloud measurements. The average aerosol particle concentration measured below cloud bases during flights AC07, AC09, AC18, and AC19 were 2417 cm$^{-3}$, 737 cm$^{-3}$, 809 cm$^{-3}$ and 428 cm$^{-3}$, respectively. b) Cloud profiling maneuvers during flight AC19 above the Atlantic Ocean near the Amazon River delta shown as three-dimensional profiles corresponding to the two dimensional profile in panel a). Relevant flight segments - particularly legs below cloud base and within cloud base, as well as cloud penetrations above cloud base - are emphasized by color-coding. c) Aerosol size distributions for each flight as used in this study.

## 2.1 Aerosol size distribution below cloud base

Aerosol size distributions were measured using an Ultra-High Sensitivity Aerosol Spectrometer (UHSAS; Droplet Measurement Technologies, Inc., Longmont, CO, USA) (Cai et al., 2008; Moore et al., 2021). The UHSAS combines a high-power infrared laser ($\lambda$ = 1054 nm) and a large solid angle range in a side-ways direction for the detection of light scattered by individual particles (Andreae et al., 2018). The aircraft instrument measures particles in the diameter size range between 92 nm and 600 nm. The instrument is mounted in an under-wing canister. The sampled air is entering the instrument by a forward facing diffusor inlet, and the airflow is reduced by a second inlet to approximately isokinetic conditions. The measured particle diameters can be assumed to be close to their dry diameters due to heating effects (Chubb et al., 2016). The UHSAS was calibrated with monodisperse polystyrene latex (PSL) spheres of known size. Typical uncertainties of UHSAS measurements are both 15 % in diameter and concentration (Cai et al., 2008; Moore et al., 2021).

The total particle number concentration in the size range of $\sim$ 10 nm to $\sim$ 500 nm ($N_{CN}$) below cloud base were measured using the Aerosol Measurement System (AMETYST); the uncertainty of these measurements is estimated to be 10 % (Andreae et al., 2018). $N_{CN}$ was measured by a butanol-based condensation particle counter (CPCs, modified Grimm CPC 5.410 by Grimm Aerosol Technik, Ainring, Germany) with a flow of 0.6 L min$^{-1}$. Particle losses in the sampling lines have been estimated and taken into account with the particle loss calculator by von der Weiden et al. (2009). Typical uncertainties of CPC measurements are on the order of $\sim$10 % (Petzold et al., 2011; Andreae et al., 2018).

The geometric mean of the aerosol size distribution and $N_{CN}$ below cloud were calculated. The mean aerosol size distribution was fitted by one-modal lognormal distributions. The integral of the fit for the aerosol size distribution should be similar to $N_{CN}$ if mainly accumulation mode particles are present. This was fulfilled for AC07, AC09 and AC18, but not for AC19 (Tables S1-S4). For this latter flight, the integrated number concentration of the monomodal lognormal fit made up approximately half of the total $N_{CN}$. This discrepancy led to the assumption that a significant number concentration of particles in the size range of Aitken mode particles were present during AC19, but not captured by the UHSAS measurements. Consequently, a bimodal ASD shape was inferred. The geometric parameters for the lognormal distribution assumed for measurements during AC19 were based on averages of bimodal aerosol size distributions measured above the ocean in previous studies (Figure S4) (Wex et al., 2016; Quinn et al., 2017; Gong et al., 2019). Other shapes of marine aerosol size distributions, e.g. as reported by Leaitch et al. (2010), were not considered for our lognormal fit because they were not in agreement with the measured UHSAS data. The resulting shape of the two modes based on literature data was weighted by the difference between UHSAS and CPC measurements (Table S4). The number concentrations of all fitted aerosol size distributions were normalized to the measured $N_{CN}$. The variability of the aerosol number size distributions was calculated by the standard deviation on average $\sim$ 10 % and up to $\sim$ 20 % for very clean conditions. As a conservative approach $\sim$ 20 % was used in our model sensitivity study to take into account the impact of this variability on cloud droplet number concentration (Section 4.2). All concentrations are reported for normalized atmospheric conditions (corrected for standard conditions (STP): T = 273.15°C and p = 1013.25 mbar).

## 2.2 Cloud droplet measurements at cloud base

Cloud droplet number concentrations and size distributions were measured by a Cloud Combination Probe – Cloud Droplet Probe (CCP-CDP) and by a Cloud and Aerosol Spectrometer with Depolarization (CAS-DPOL) mounted onboard HALO (Wendisch et al., 2016; Voigt et al., 2017). Cloud droplet number size distributions (DSDs) between 3 $\mu$m and 50 $\mu$m in diameter were measured at a temporal resolution of 1 s by the CAS-DPOL and CCP–CDP probes (Baumgardner et al., 2011; Voigt et al., 2010, 2011; Kleine et al., 2018; Wendisch and Brenguier, 2013). These probes have different measurement characteristics such as particle inlet, sampling area of detection, size sensitivities etc. The CCP-CDP is an open-path instrument that detects forward-scattered laser light from cloud particles as they pass through the CDP detection area (Lance et al., 2010). CAS-DPOL collects forward-scattered light to determine particle size and number that pass the sampling area centered in an inlet shaft that guides the airflow. CCP-CDP and CAS-DPOL has similar values of uncertainty ($\sim$ 10%) in the sample area. However, particle velocities in the sampling tube may be modified by the CAS tube when compared to the open path instruments (like CCP-CDP). This results in an additional uncertainty in the droplet number concentration measured by CAS-DPOL. During the ACRIDICON-CHUVA campaign the resulting uncertainty in the droplet concentration measured by CCP-CDP and CAS-DPOL were $\sim$ 10% and $\sim$ 21 %, respectively (Braga et al., 2017a).

For cloud base measurements, each probe DSD spectrum represented 1 s of flight path (covering between 70 m to 120 m of horizontal distance for the aircraft speed at cloud bases). We refer in the current study to the measurements closest to cloud base as 'cloud base' measurements, even if the actual cloud base might have been slightly below this altitude of measurements (Section 3.2.2 and Figure 2). The cloud base measurements were selected based on the videos recorded by the HALO cockpit forward-looking camera. From the DSDs, the droplet number concentrations were derived by size integration. Braga et al. (2017a) showed that both probes were in agreement within their uncertainty range for probe DSDs ($\pm \sim 16\%$). The overall systematic errors in the cloud probe integrated water content with respect to a King type hot-wire device are $\sim$ 6 % for CAS-DPOL and $\sim$ 21 % for CCP-CDP. A positive bias of $\sim 20\%$ was found for CAS-DPOL droplet concentration in comparison with those measured with CCP-CDP for cloud passes with cloud droplet effective radius < 7 $\mu$m (mostly measured at cloud bases). Cloud passes were defined for conditions, under which the number droplet concentration (i.e., particles with diameter larger than 3 $\mu$m) exceeded 20 cm$^{-3}$. This criterion was applied to avoid cloud passes well mixed with subsaturated environment air (RH < 100%) and counts of haze particles, typically found at cloud edges. Additional details about the cloud probes measurements at cloud bases used in this study can be found in Tables S5-S6.

## 3 Methodology

### 3.1 Probability matching method (PMM): Pairing measured updraft velocities ($w$) and droplet number concentrations ($N_{d,m}$)

The thermal instability in the boundary layer promotes the formation of clouds consisting of regions with updrafts and downdrafts. At cloud bases, the variability in vertical velocities and droplet concentration is high due to air turbulence. Clouds

develop in updrafts, and during their vertical development the continued movement as a turbulent eddy adds a large random component to the relationship of $w$ with $N_{d,m}$. These intrinsic characteristics of clouds reduce the confidence that a measured $w$ in the cloud led to the simultaneously measured $N_{d,m}$. Such inconsistencies often result in poor correlations of $w$ and $N_{d,m}$. As $w$ is highly variable in clouds and is measured independently from $N_{d,m}$, we apply the "Probability Matching Method" (PMM, Haddad and Rosenfeld (1997)) to statistically determine the most probable combinations of $N_{d,m}$ and $w$ values using the same percentiles of their occurrence. The PMM analysis is based on the assumption that these two related variables increase monotonically with each other. This assumption implies that entrainment – which may lead to a reversal of the assumed trend - can be neglected near cloud base which is likely a valid assumption under these conditions. Measured $N_{d,m}$ and $w$ values were sorted in ascending order and the most likely $w$ value was assigned to $N_{d,m}$ for each of the four flights. To avoid biases caused by outlier measurements, $N_{d,m}$ above the 97.5[th] and below the 2.5[th] percentile were removed. Only cloud passes with positive vertical velocities (i.e., updrafts) were considered in the analysis. Furthermore, we take into account only data of non-precipitating clouds, typically from cumulus humilis and cumulus mediocris clouds. Braga et al. (2017a) have shown that the PMM can be used to find the best agreement between measured and estimated $N_d$ at cloud base as a function of $w$. In the current study, PMM analysis is used to compare the $N_{d,m}$ at cloud base and its assigned $w$ with $N_{d,p}$ at a constant $w$ in the model. Figure 2 shows an example of measured $w$ at cloud bases and estimated $w$ based on the PMM analysis ($w_{PMM}$). The figure shows that $w_{PMM}$ are in well agreement with measurements at cloud bases. Furthermore, for cloud passes in which the values of $w$ are negative realistic $w$ are estimated based on PMM.

## 3.2 Adiabatic cloud parcel model

### 3.2.1 Model description and simulations

The adiabatic parcel model describes the growth of aerosol particles by water vapor uptake on a moving mass grid (Feingold and Heymsfield, 1992; Ervens et al., 2005). The air parcel is described to rise with a constant $w$ below and inside of the cloud. Saturation with respect to water vapor in the air parcel is calculated based on the standard thermodynamic equations for adiabatic conditions as a function of $w$ and particle properties ($N_a$, particle sizes and hygroscopicity) (Pruppacher and Klett, 1997). It is assumed that the aerosol particles are internally mixed with identical hygroscopicity ($\kappa$) of all particles. This assumption was made based on previous sensitivity studies that have shown that for marine and aged continental air masses internal mixtures are suitable approximations (Ervens et al., 2010). We note that $\kappa$ is regarded here as an effective parameter, encompassing all factors that affect water uptake. Simulations are performed up to a height of 70 m above the level of predicted maximum supersaturation. The initial conditions for the model simulations are summarized in Tables S1-S4. Particles that exceed a diameter of 3 $\mu$m are defined as droplets; this definition allows a direct comparison of $N_{d,p}$ and $N_{d,m}$. Collision/coalescence processes are not considered as we restrict our analysis to heights near cloud base where droplets are relatively small and the cloud droplet size distribution is narrow. Under such conditions, collision-coalescence is likely negligible (Shaw et al., 1998; Xue et al., 2008; Rosenfeld, 2018; Braga et al., 2017b). Sensitivity studies are performed for the

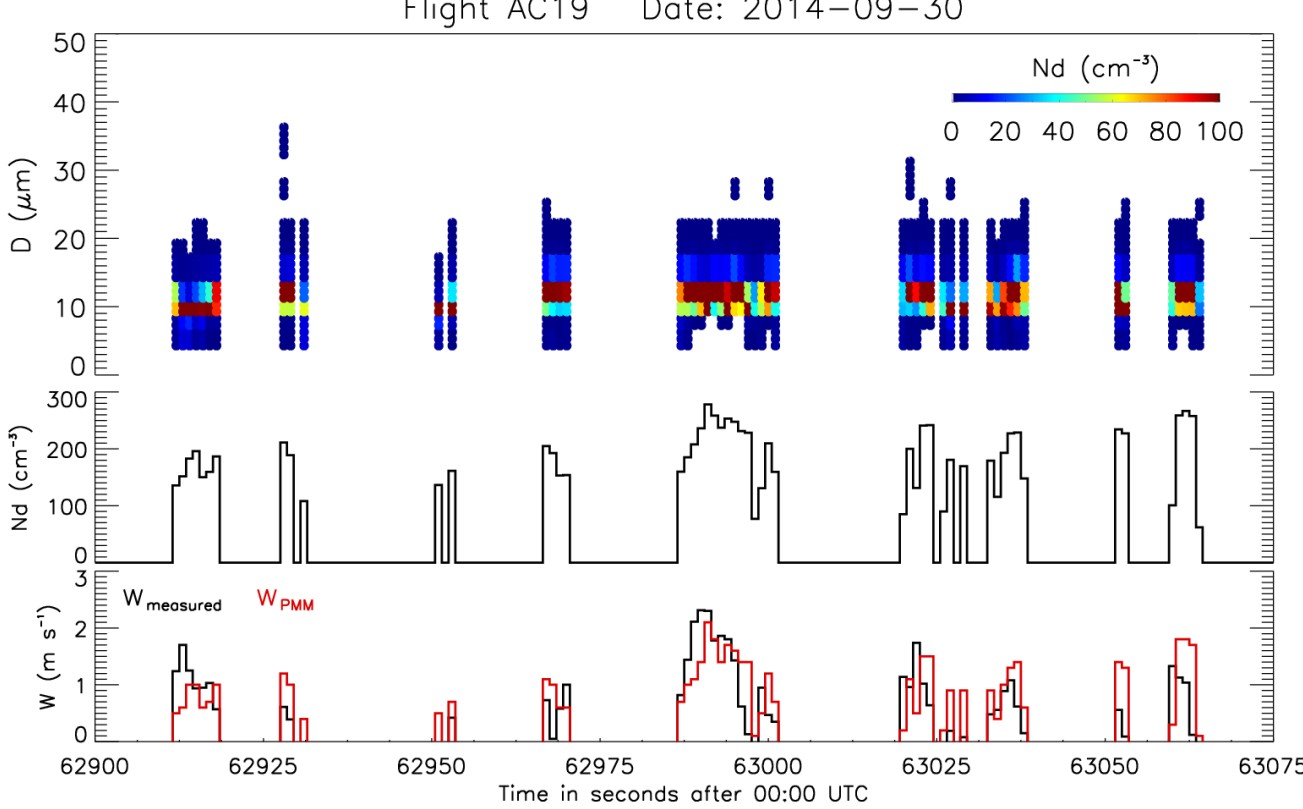

**Figure 2.** Time series of droplet size distribution measured by CCP-CDP [top], number concentration of droplets ($N_d$) [middle], measured vertical velocities $w$, and estimated $w$ based on PMM method [bottom]. The measurements were performed during flight AC19 above the Atlantic Ocean.

observed ranges of $w$ and $N_a \pm 40\%$ and assumed range of $0.02 \leq \kappa \leq 1$ to identify parameter ranges and combinations for which droplet closure can be achieved.

### 3.2.2 Determination of in-cloud height to compare $N_{d,m}$ and $N_{d,p}$

The cloud base measurements were performed at approximately constant altitude during each research flight and were selected based on the videos recorded by the HALO cockpit forward-looking camera. However, these measurements might represent different levels in relation to the level of maximum supersaturation at cloud bases, which depends on the updraft velocity and turbulence in cloud. In order to determine the height at which $N_{d,m}$ and $N_{d,p}$ should be compared, the measured liquid water content (LWC) was compared to the simulated LWC using the aerosol size distribution for the different flights together with $w$ measured at cloud base and assumed hygroscopicity of $\kappa = 0.1$.

Under adiabatic conditions, $N_{d,p}$ is predicted to be approximately constant at $\sim 20$ m above the level of the maximum supersaturation $S_{max}$ (Fig. S5). Figure 3 shows the values of predicted LWC based on the simulations as a function of height above $S_{max}$ for the four flights. Overlaid on the model results (colored lines) are the frequencies of measured LWC by the cloud probes near cloud base (white bars). The measured LWC represents the cumulative mass size distribution. For all flights, the model predictions in most of cases match the minimum LWC measured at $\sim 20$ m above the $S_{max}$ level. This height level might

represent slightly different absolute heights above the surface and the level of saturation estimated by the model (RH = 100%) (Fig. S6). However, since we focus our discussion in the following section on the comparison of $N_{d,m}$ and $N_{d,p}$, we perform our analysis based on model predictions at a height of 20 m above $S_{max}$.

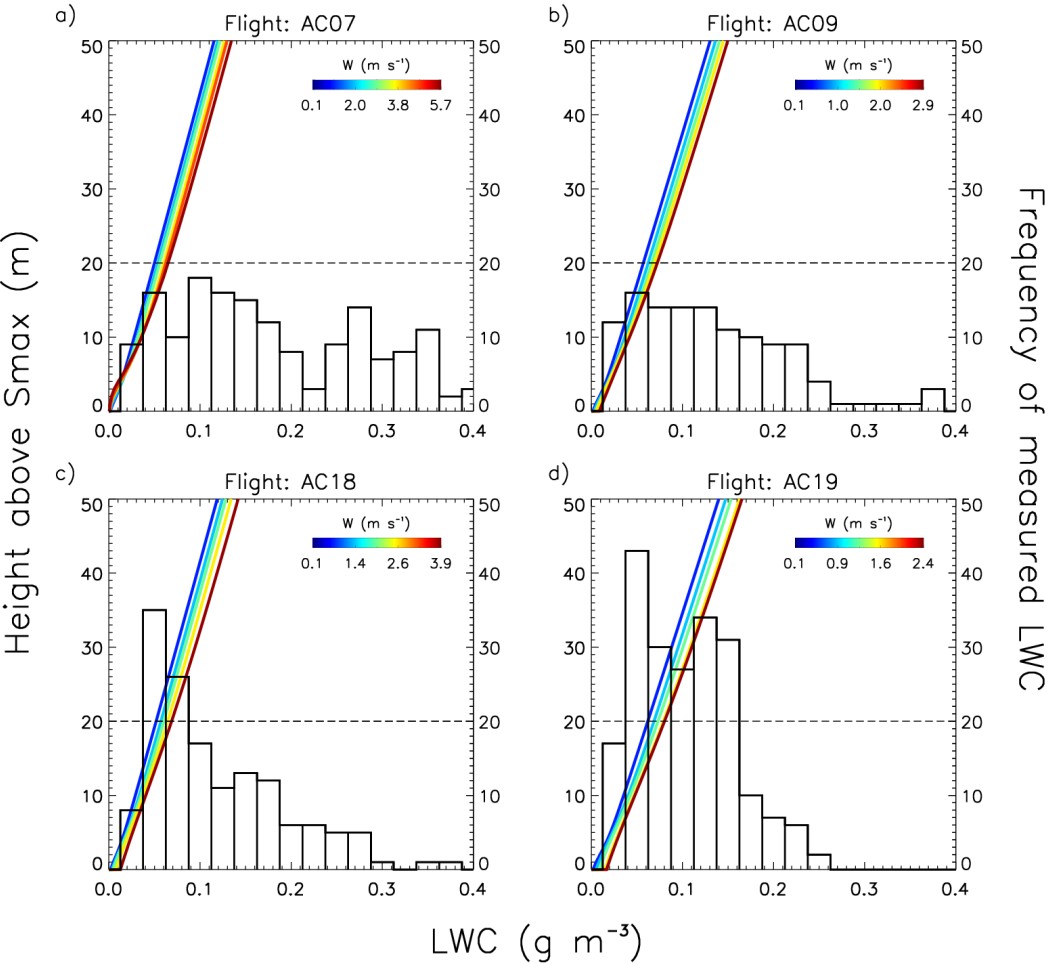

**Figure 3.** Predicted LWC [g m$^{-3}$] as a function of height above the level of $S_{max}$ [left axis] and $w$ (lines, color-coded by $w$ [m s$^{-1}$]) for flights a) AC07, b) AC09, c) AC18, d) AC19. The vertical bars indicate the number of cloud passes (with a temporal resolution of 1 s) as a function of the measured LWC by CAS-DPOL and CCP-CDP in cloud (right axis). The dashed line denotes the level of 20 m above predicted maximum supersaturation, at which $N_{d,p}$ is predicted (Section 3.2.2).

## 4 Results and Discussion

### 4.1 Constraining aerosol hygroscopicity ($\kappa$) based on $N_d$ and $w$

Figure 4 shows the range of $N_{d,m}$ as a function of $w$ as determined by the PMM (Section 3.1); the symbols indicate $N_{d,m}$ from CAS-DPOL (black diamonds) and CCP-CDP (black triangles). The lines in Fig. 4a-d represent model predictions for the assumption of $\kappa = 0.05$, $\kappa = 0.1$, $\kappa = 0.3$ and $\kappa = 0.6$ for up to 38 $w$ values for each flight, covering the measured $w$ range. Figure 4e shows results of additional simulations for Flight AC19 (marine conditions) assuming $\kappa = 0.6$ and $\kappa = 0.8$ for aerosol particles from Aitken ($d < 70$ nm) and $\kappa = 0.1$ and $\kappa = 0.2$ for aerosol particles from accumulation ($d \geq 70$ nm) mode sizes, respectively. For all flights, $N_{d,m}$ values are reasonably reproduced by the model assuming a particle hygroscopicity of $0.05 < \kappa < 0.3$; $N_{d,p}$ are closer to the measured values from CCP-CDP assuming a slightly lower $\kappa$, whereas $N_{d,m}$ from CAS-DPOL indicate a slightly higher $\kappa$. However, these deviations are within the uncertainty range of the cloud probe measurements, i.e., $\sim 10\%$ and $\sim 21\%$ for CCP-CDP and CAS-DPOL, respectively (Braga et al., 2017a).

Figure 4 shows that the agreement between measured and predicted cloud droplet number concentration is obtained for low $w$ during all flights. However, the value of $w$, above which the model predictions deviate from measurements varies among the flights: For continental clouds as encountered during AC07, AC18 and AC09, the model results agree well with observations for $w \lesssim 2.5$ m s$^{-1}$. At higher $w$, $N_{d,m}$ shows a much stronger increase with $w$ than predicted by the model. For AC19, i.e., above the ocean, this trend is even obvious for $w \gtrsim 0.5$ m s$^{-1}$. The statistical analysis based on bias, root mean square error (RMSE), and mean absolute error (MAE) from the closure analysis are shown in Tables S7-S18. This analysis suggests that the use of two probes to perform the closure does not have a large effect on the inferred value of $\kappa$. We find best agreement, quantified by the smallest absolute bias and RMSE, for all cases for single $\kappa$ values of $0.05 \leq \kappa \leq 0.2$. The deviations between $N_{d,m}$ from CCP-CDP and CAS-DPOL ($\sim 21\%$ on average) reinforce the advantage of duplicate measurements for the closure analysis. The use of a single cloud probe might lead to a biased $\kappa$ estimate based on the data set of each cloud probe separately. The consideration of both cloud probes shows the uncertainty in $N_d$ measurements and therefore the uncertainty range of $\kappa$ and/or $N_a$ values for $N_d$ closure. Therefore, we base our conclusions in the following on the statistical analysis of all data from both probes together (Tables S9, S12, S15 and S18).

The results in Fig. 4 imply that the assumption of an internally mixed aerosol population with moderate hygroscopicity ($\kappa \sim 0.1$) is justified to reproduce $N_{d,m}$ for flights AC07, AC09 and AC18 for wide ranges of updraft speeds (0.1 m s$^{-1} \leq w \leq$ 2.5 m s$^{-1}$). This $\kappa$ value has been suggested previously for comparable air masses during the dry season in the Amazon Basin (e.g., Pöhlker et al., 2016, 2018). In these prior studies, $\kappa$ was constrained based on size-resolved CCN measurements and measurements of the aerosol chemical composition, dominated by an aged organic fraction. Our results are in agreement with previous studies. The value range is representative of internally mixed aerosol particle populations during the dry season in the Amazon Basin, which are influenced by fresh and aged biomass burning aerosol from Amazon and Africa.

While also particles of different hygroscopicities and activation thresholds depending on $w$ might explain the trends in Fig. 4a-c, there is no indication of higher hygroscopicity of smaller accumulation mode aerosol particles during the Amazonian dry season (e.g., Pöhlker et al., 2016, 2018). In air masses of different origin, aerosol particles would likely not only exhibit different

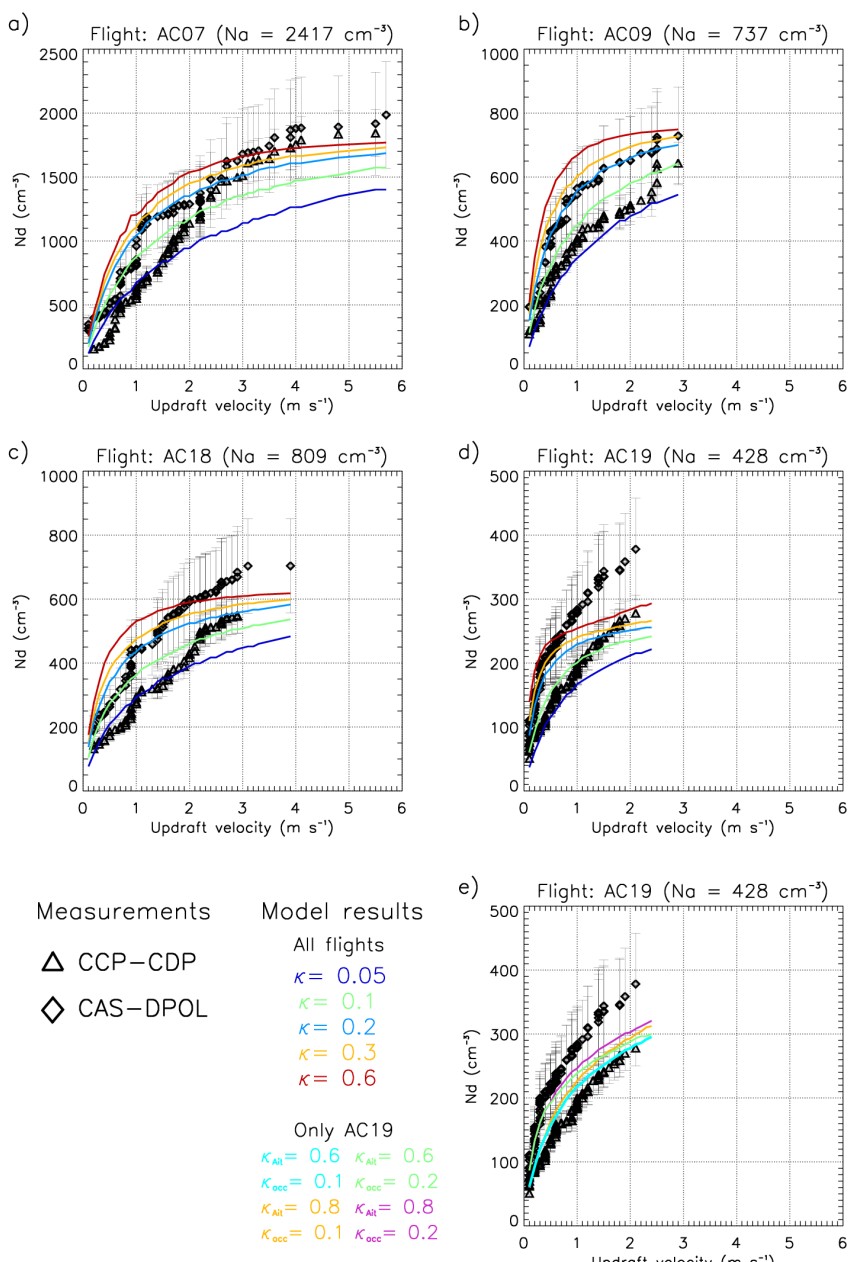

**Figure 4.** Cloud droplet number concentration ($N_d$) as a function of updraft velocity near cloud base of convective clouds during flights: a) AC07, b) AC09, c) AC18, d) and e) AC19. The measured updraft velocities are based on the "probability matching method" (PMM) using the same percentiles for updraft velocity and $N_{d,m}$ (Section 3.1). The black diamonds and triangles represent $N_{d,m}$ near cloud base from the CAS-DPOL and CCP-CDP probes, respectively. Measurement uncertainties, indicated by error bars, are $\sim 21\%$ and $\sim 10\%$ for CAS-DPOL and CCP-CDP data (Braga et al. (2017a)). The colored lines in panels a) - d) show $N_{d,p}$ assuming a single $\kappa$ value for both modes (labeled on the left). Panel e) shows $N_{d,p}$ based on simulations assuming different values of $\kappa$ for Aitken and accumulation mode particles during flight AC19.

chemical composition and hygroscopicity but also large variability in their particle number concentrations. Given the relatively small standard deviations in the measured $N_a$ (Tables S1 - S4), we are confident that the sampled aerosol populations did not have large variability in their composition. The chemical composition of Aitken mode particles often differs significantly from that of accumulation mode particles, which are more aged and internally mixed (e.g., (Wex et al., 2016; Pöhlker et al., 2018)), and thus continental Aitken mode particles usually exhibit a lower hygroscopicity than accumulation mode particles (McFiggans et al., 2006).

The air masses below cloud encountered during flight AC19 were mostly impacted by marine air, as supported by prior back trajectory analysis (Section S1 and Holanda et al. (2020)) and exhibited a bi-modal aerosol size distribution with low $N_{d,m}$ (Figure 1c). For this flight, the cloud droplet closure is worse as compared to the reasonable agreement for the other three cases. Not only is the absolute difference between $N_{d,m}$ and $N_{d,p}$ relatively larger (Figure 4d), but also the trend of $N_{d,m}$ with $w$ cannot be well reproduced: While at $w < 0.5$ m s$^{-1}$, the range of $N_{d,p}$ agrees well with $N_{d,m}$, above this threshold the model strongly underestimates the droplet number concentration even for $\kappa = 0.3$ (Figure 4d). Assuming $\kappa = 0.6$ only slightly increases $N_{d,p}$ as compared to the results for $\kappa = 0.3$. This trend shows that $N_{d,p}$ is rather insensitive to $\kappa$ if particles are very hygroscopic ($\kappa \gtrsim$ 0.3). While all $\kappa$ values lead to reasonable agreement at low $w$, none of the model results can reproduce the strongly increased $N_{d,m}$ with $w$. Therefore, we conclude that the simplifying assumptions made in the model, i.e., identical hygroscopicities across both aerosol modes, may not be appropriate.

The measured aerosol size distribution during flights AC19 differed significantly from the other ones (Figure 1c) because of (i) low $N_a$, and (ii) a distinct Aitken mode (mean diameter 37 nm) that comprised $\sim 47\%$ of the particle number concentration. At such low $N_a$, the maximum supersaturation in the clouds is relatively high so that at sufficiently high $w$, Aitken mode particles (diameter $\lesssim 70$ nm) may be activated into cloud droplets and contribute to $N_d$ (Pöhlker et al., 2021). Highly hygroscopic Aitken mode particles over the ocean may reflect secondary marine sulfate aerosols (Andreae and Raemdonck, 1983).

To account for different hygroscopicities in Aitken and accumulation modes, we performed further sensitivity analyses using combinations of $\kappa = 0.1$ and 0.6 for the two modes (Figure 4e). It is obvious that the choice of $\kappa$ for the Aitken mode ($\kappa_{\mathrm{Ait}}$) does not affect $N_{d,p}$ for $w \lesssim 1$ m s$^{-1}$ in the presence of very hygroscopic accumulation mode particles ($\kappa_{\mathrm{acc}} = 0.6$) or below $w$ $\lesssim 0.5$ m s$^{-1}$ with $\kappa_{\mathrm{acc}} = 0.1$, respectively. Even assuming rather extreme values of $\kappa_{Ait} = 0.8$ cannot fully reproduce the large increase in $N_d$ at $w \gtrsim 1.5$ m s$^{-1}$ as observed by the CAS probes; assuming very hygroscopic Aitken mode and less hygroscopic accumulation mode particles can approximately reproduce the trend in $N_{d,m}$ from the CDP. It should be kept in mind that $\kappa$ is considered an effective parameter that may also reflect water uptake due to additional processes or effects that are not represented in our model and therefore cannot be further reconciled here.

Varying $\kappa_{\mathrm{acc}}$ from 0.1 to 0.6 leads to a large increase of $N_{d,p}$ at all $w$. The corresponding change in $N_{d,p}$ by increasing $\kappa_{\mathrm{Ait}}$ is much smaller. The reason for this relatively smaller sensitivity of $N_{d,p}$ to $\kappa_{\mathrm{Ait}}$ is the fact that the supersaturation in the cloud is mostly controlled by the droplet growth on accumulation mode particles. The sensitivity of $N_{d,p}$ formed on Aitken mode particles to $\kappa_{\mathrm{acc}}$ is slightly larger if $\kappa_{\mathrm{acc}} = 0.1$ as compared to $\kappa_{\mathrm{acc}} = 0.6$, because in the latter case the supersaturation is efficiently suppressed preventing a higher number of Aitken mode particles from activating. Overall we can conclude that assuming different $\kappa$ values for accumulation and Aitken mode leads to a better representation of the observed trends of $N_{d,m}$

with $w$ (Tables S16 and S17). However, in the absence of more information on the particle hygroscopicity we cannot state with certainty that the assumptions of the two $\kappa$ values are appropriate for this aerosol population. Figure 4d clearly shows that the simplified assumption of a single $\kappa$ is not appropriate to infer $N_{d,p}$ for low aerosol loading and when the particle number concentrations of the accumulation and Aitken modes are comparable. By using a single $\kappa$ value, we cannot reproduce the observed continuously strong increase of $N_{d,m}$ for the whole $w$ range. Instead we predict a smaller increase at $w \sim 1$ m s$^{-1}$, i.e., a flattening of the curve.

In general, the observed trends of $N_d$ with $w$ for flights AC07, AC09 and AC18 confirm results from previous sensitivity studies that have shown that with increasing $w$, changes in $N_d$ become small and, thus, sensitivity of $N_d$ to $\kappa$ and $w$ decreases (Ervens et al., 2005; Fountoukis et al., 2007; Reutter et al., 2009). In these studies, it was demonstrated that the sensitivity of $N_d$ to $N_a$ becomes small when nearly all particles are activated ('aerosol-limited regime'). For these simulations, either only an accumulation mode was considered, or $N_d$ closure studies were performed for situations with low $w$ and/or fairly small Aitken mode particles (< 40 nm) that were not predicted to activate.

Anttila and Kerminen (2007) showed in a model study focusing only on Aitken mode particles that $N_d$ is highly sensitive to the chemical composition of Aitken mode particles. In our recent model study, we systematically explored the extent to which the presence of an Aitken mode might significantly affect $N_d$ as a function of updraft velocity (Pöhlker et al., 2021). In that study, we show that the sensitivities of $N_{d,p}$ are different to the properties ($N_a$, $\kappa$) of accumulation and Aitken mode particles, respectively. Generally, we find that $N_{d,m}$ is not highly sensitive to Aitken mode particle properties in the presence of a dominant accumulation mode, which is in agreement to our results in Figures 4 and S7.

## 4.2  Influence of aerosol number concentration ($N_a$) on predicted $N_d$

The measurements of $N_a$ were associated with uncertainties of $\pm \sim 20\,\%$ (Section 2.1). In order to account for this uncertainty and possible fluctuation in $N_a$ at cloud base, $N_{d,m}$ and $N_{d,p}$ are compared for all flights, using $N_a$ (Figure 1), reduced and increased by 20%, 30% and 40% as model input, respectively. Figure 5 shows the comparison of measured and predicted $N_d$ assuming the uncertainty range of $N_a$. Figure 4a-d shows the $N_{d,p}$ with the values of $\kappa$ that are within the uncertainty range of cloud probes measurements. The green lines show the model results for $\kappa = 0.1$ for flights AC07, AC09 and AC18 and $\kappa = 0.2$ for flight AC19, which show the smallest absolute bias and RMSE to the measured data (Tables S9, S12, S15 and S18); the other lines denote $N_{d,p}$ using the higher and lower input $N_a$. For flights AC07, AC09 and AC18, $N_{d,p}$ is within the range of $N_{d,m}$ for the assumed model parameter space. Also the curves for $N_{d,p}$ as a function of $w$ exhibit a similar shape as predicted for a variation in $\kappa$. The agreement between measurements and model results decreases with increasing $w$. However, unlike the $N_{d,p}$ curves for high $\kappa$ that level off at high $w$ when all particles are activated, an increase in $N_a$ leads to continuously higher $N_d$ as the aerosol-limited regime is not yet reached.

Using different $\kappa$ values for the two modes leads to a better representation of the $N_d$ trend with $w$, i.e., the shape of the curve can be fairly reproduced for different combinations of separate values of $\kappa_{acc}$ and $\kappa_{Ait}$. However, $N_d$ is systematically underestimated which suggests that in addition uncertainties in $N_a$ are important for $N_d$ closure and likely more important than those in $\kappa$. In fact, the closure results in Fig. 5 show that $N_{d,m}$ can be reproduced well by the model over wide $w$ ranges

if the variability in $N_a$ of $\pm20\%$ and the uncertainty in $N_{d,m}$ is taken into account, and an average hygroscopicity of $\kappa = 0.1$ is assumed for the aerosol in the Amazon basin during the dry season and $\kappa = 0.2$ for that in the western tropical Atlantic. However, also the assumption of two different $\kappa$ values leads to good $N_d$ closure for the bimodal ASD as observed during flight AC19 (Fig. 5e). These two different assumptions on the mode hygroscopicities result in ambiguous conclusions on the importance of knowledge of $\kappa$ values for Aitken mode particles contribute to $N_d$ (Pöhlker et al., 2021). Generally, for both sets of simulations, i.e., varying $N_a$ or $\kappa$, the agreement between model and measurements is best at low $w$. At $w \gtrsim 2\,\mathrm{m\,s^{-1}}$, the $N_{d,p}$ curves flatten suggesting a decreasing sensitivity of $N_{d,p}$ above this $w$ threshold. Unlike the trends in $N_{d,p}$ for different $\kappa$ values, the lines for different $N_a$ keep diverging with increasing $w$. Thus, the sensitivity of $N_{d,p}$ to $N_a$ is predicted to remain high, and nearly independent of $w$ which may point to conditions near the aerosol-limited regime.

## 4.3 Sensitivities of $N_d$ predictions to $w$, $N_a$ and $\kappa$

The sensitivities of cloud drop number concentrations to hygroscopicity ($\kappa$), $N_a$ and $w$ have been explored in numerous previous studies. In the following, we place our results in the context of such studies. Consistent with such previous sensitivity studies, we calculated the sensitivity $\xi$ of $N_{d,p}$ to $\kappa$, $N_a$ and $w$

$$\xi(X) = \frac{\partial ln N_d}{\partial ln X} \tag{E.1}$$

whereas X is $\kappa$, $N_a$ or $w$, respectively. The results are summarized in Figure S7. They show that $\xi(\kappa)$ is smallest as compared to $\xi(N_a)$ and $\xi(w)$. $\xi(\kappa)$ is highest for low $\kappa$ as conditions, the activated fraction is smallest and thus a small change in $\kappa$ might cause a significant change in $N_d$. $\kappa$ is often termed 'effective hygroscopcity' since it is used as a parameter that reflects multiple parameters that affect the water uptake by the particles. Generally, sensitivities are high under conditions of high supersaturation which are present at high $w$ and/or low $N_a$. While the sensitivities calculated for flights AC07, AC09 and AC18 follow these general trends according to $N_a$, the $\xi(X)$ values are higher for AC19 even though $N_a$ was lowest for this flight. The reason for this difference is the successive activation of Aitken mode particles at high $w$ (Pöhlker et al., 2021). A sensitivity study for the same flights has been performed previously (Cecchini et al., 2017) in which the sensitivities of $N_d$ and effective cloud droplet diameter to $N_a$ and $w$ were explored in detail at various heights in cloud. The focus of that study was the change of the sensitivities during cloud evolution, i.e., as a function of height in cloud. In the present study, we focus on the sensitivities of $N_d$ near cloud base, but additionally explore the importance of $\kappa$ in determining $N_d$. Such analysis can be used to give guidance for future measurements in similar clouds on the absolute values and relative importance of the three parameters to predict $N_d$.

Generally, prior sensitivity studies agree in the rankings of the relative importance of hygroscopicity ($\kappa$), $N_a$ and $w$, as also shown in Figure S7. Feingold (2003) has shown that $N_a$ has the largest influence on effective radius which is indirectly related to $N_d$. The sensitivities to the effective radius are typically smaller than those to $N_d$ (Pardo et al., 2019). In our recent model study, we have shown that in the transitional regime, i.e., in the parameter space between the aerosol- and updraft limited regimes, as defined by Reutter et al. (2009), $N_d$ can be equally sensitive to $\kappa$ and $w$ (Pöhlker et al., 2021). In that latter study, it was shown that with increasing $N_a$, the sensitivities to both parameters decrease; however, the sensitivity of $N_d$ to $w$ remains higher under such conditions than that to $\kappa$.

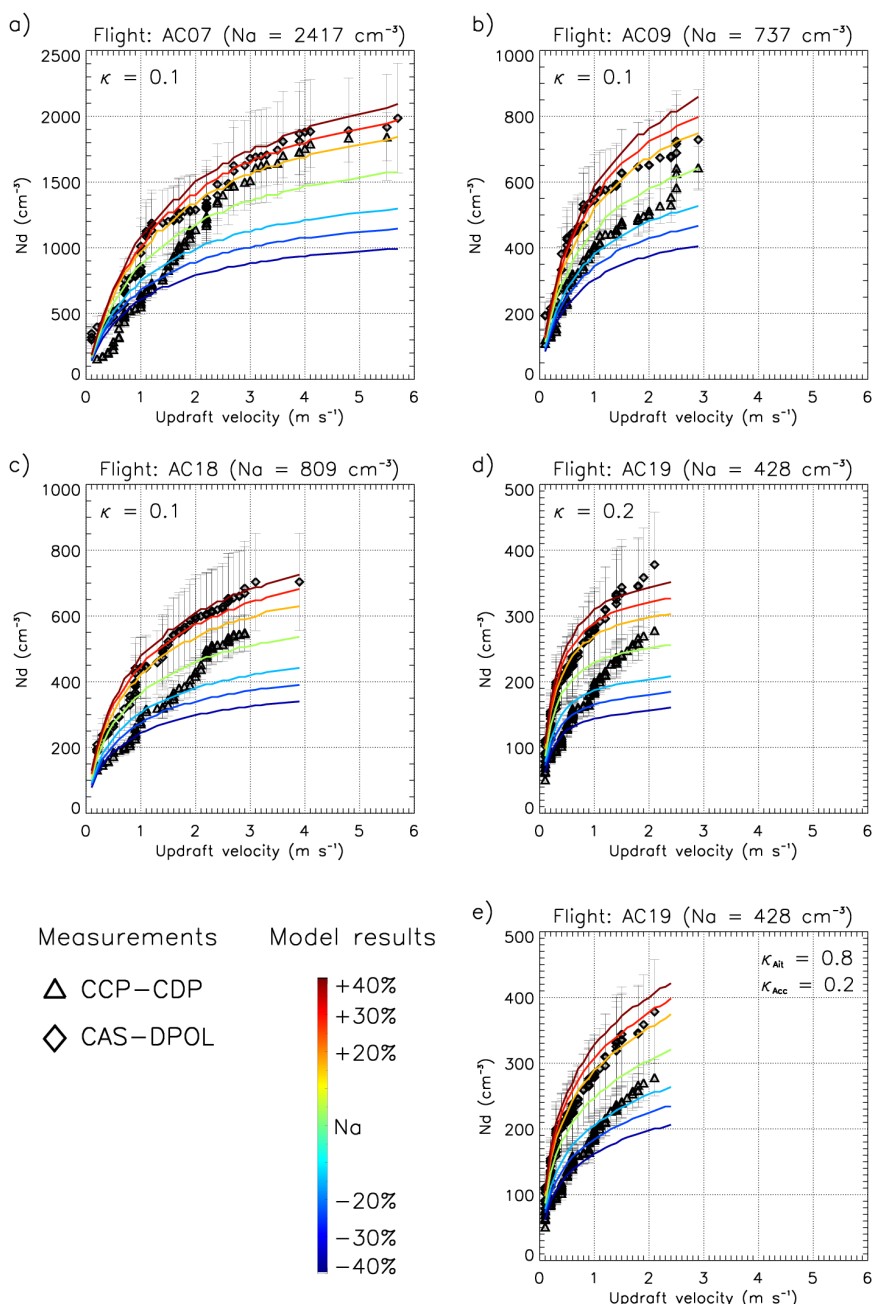

**Figure 5.** Cloud droplet number concentration ($N_d$) as a function of updraft velocity near cloud base of convective clouds during flights: a) AC07, b) AC09, c) AC18, d) and e) AC19. The measured updraft velocities are based on the "probability matching method" (PMM) using the same percentiles for updraft velocity and $N_{d,m}$ (Section 3.1). The black diamond and triangle symbols represent $N_{d,m}$ near cloud base with the CAS-DPOL and CCP-CDP probes, respectively. Measurement uncertainties (indicated by error bars) are $\sim 21\%$ and $\sim 10\%$ for CAS-DPOL and CCP-CDP data (Braga et al. (2017a)). The lines show $N_{d,p}$ assuming the uncertainty range of $N_a$ measurements, colored-coded by $\Delta N_a$ [%].

The uncertainties in updraft measurements are larger than those of hygroscopicity due to the great variability of $w$ near cloud base. Peng et al. (2005) compared $N_{d,p}$ based on a $w$ distribution in a range of 0.09 - 1 m s$^{-1}$ and using characteristic single $w$ values. They found differences in $N_{d,p}$ on the order of < 10% for the two sets of model simulations. Meskhidze et al. (2005) performed model simulations of low-level cumuliform clouds for which a range of 0.9 m s$^{-1}$ $\leq w \leq$ 2.8 m s$^{-1}$ had been observed. They concluded that parameterizations of $N_{d,p}$ should include a weighting factor for high values of $w$ as otherwise $N_d$ might be biased high due to enhanced vertical velocity within in cloud cores as compared to cloud base.

In turbulent clouds with high $w$, the determination of $w$ near cloud bases might be challenging; however, the resulting uncertainties in updraft velocity or its distributions cannot explain the discrepancies between $N_{d,m}$ and $N_{d,p}$ at high $w$ (Figures 4 and 5). Under such conditions, the activated fraction approaches unity and any increase in $w$ would not lead to higher $N_d$ and improve the overall $N_d$ closure (e.g., Hsieh et al. (2009)). Therefore under such condition, $\xi(w)$ becomes small (Figure S7b, e, h, k) These previous $N_d$ sensitivity and closure studies either considered $w$ as a fitting parameter to obtain good closure or used $w$ values or distributions as relatively poorly constrained parameters. The PMM analysis as applied in the current study partially overcomes these uncertainties as it provides a stronger constraint of the $w$ and $N_d$ pairs for the full $w$ range (Section 3.1), as opposed to the previous studies that derived their $w$ distributions from averaging measured updraft velocities without sorting $w$ and $N_{d,m}$ data based on their frequency occurrence.

Reutter et al. (2009) termed conditions under which nearly all particles are activated into cloud droplets as 'aerosol-limited regime' when $N_d$ is only dependent on $N_a$, and not on $w$. Such conditions are present at relatively low total $N_a$ and high $w$, i.e., when the maximum supersaturation in the cloud is relatively high. When an increase in $N_a$ results in an equal increase in $N_d$, $\xi(N_a)$ approaches unity (Figure S7c, f, i, l). The measured and predicted activated fractions for flights AC07, AC09 and AC18 reach $\geq$80% at updraft velocities of $w \gtrsim 1$ m s$^{-1}$ if the measured value is based on the CAS data (Figure 4). Therefore, we conclude that the sensitivity of $N_d$ to $N_a$ is much greater than that to $w$ under these conditions which is also reflected by the rather small increase in $N_d$ with $w$ at high updraft velocities.

Overall, the variability of predicted $N_d$ due to inferred $\kappa$ ranges in the present study confirm trends from previous sensitivity studies for mono-modal aerosol size distributions: The sensitivity to $N_d$ decreases with increasing $w$, i.e. when the activated fraction is large and activation of additional smaller particles increases $N_d$ only to a small extent (Figure S7 and Ervens et al. (2005); Reutter et al. (2009); Cecchini et al. (2017); Pardo et al. (2019)). If low hygroscopicity limits the water vapor uptake, a small change in $\kappa$ may lead to a significant change in $N_d$, resulting in high $\xi(\kappa)$ values. A change of $\kappa$ by the same factor for highly hygroscopic particles, however, might not lead to a significant change in $N_d$ due to the regulation of the supersaturation ('buffering'), i.e., the efficient growth of more cloud droplets which, in turn, reduces the supersaturation. Our sensitivity study of AC19 exceeds these previous sensitivity studies that focused on monomodal aerosol size distributions. We show that the uncertainties in $N_{d,p}$ become larger under conditions when Aitken mode particles contribute to $N_d$ as only at very high $w$ the aerosol-limited regime is reached and $\xi(\kappa)$ and $\xi(w)$ decrease. Qualitatively this was also suggested in a previous $N_d$ closure study for marine stratocumulus clouds, where it was concluded that only the presence of an Aitken mode could explain the high $N_{d,m}$ at updraft velocities of $w \geq 1$ m s$^{-1}$ (Schulze et al., 2020). Generally, the conditions at which Aitken mode particles contribute to CCN depend on the combinations of the parameter values of $N_a$, $w$ and $\kappa$ (Pöhlker et al., 2021). Therefore, Aitken

mode particles were shown to contribute to CCN in Arctic stratocumulus clouds or fog, that are characterized by low $w$ and $N_a$ (Korhonen et al., 2008; Leaitch et al., 2016; Jung et al., 2018), whereas both updraft and aerosol loading are much higher in the convective cumulus clouds in the Amazon. Our analysis exceeds these studies as we show that the $w$ threshold above which Aitken mode particles contribute to $N_d$ depends on the properties (e.g., $\kappa$, $N_{acc}$) of the accumulation mode. In addition, we show that various combinations of inferred $\kappa_{acc}$ and $\kappa_{Ait}$ result in similar $N_{d,p}$ and thus cannot be constrained without more detailed composition measurements. While these conclusions are drawn on a single observationally-based case study, a more systematic analysis of parameter ranges of Aitken and accumulation mode particles is provided in our recent study (Pöhlker et al., 2021).

## 5   Summary and conclusions

Airborne measurements of cloud droplet number concentrations ($N_{d,m}$), aerosol particle size distributions and updraft velocities ($w$) near cloud base were performed during the ACRIDICON-CHUVA campaign in September 2014. Using an adiabatic air parcel model, the importance of aerosol particle number concentration ($N_a$) and effective hygroscopicity ($\kappa$) and their uncertainties on predicted cloud droplet number concentrations ($N_{d,p}$) near cloud bases of growing convective cumuli, formed over the Amazon and western Atlantic, were explored. Data from aerosol and cloud probes onboard HALO were used as model input for this cloud droplet closure analysis. Model results for four different scenarios in terms of aerosol loading and size distributions, and of $w$ confirm previously suggested values of the hygroscopicity parameter $\kappa$ to reasonably predict $N_d$ for most conditions: best $N_d$ closure is achieved for an effective hygroscopicity of $\kappa \sim 0.1$ for the Amazon basin during the dry season using the full data set of CCP-CDP and CAS-DPOL measurements. Above the western Atlantic best $N_d$ closure was achieved for $\kappa \sim 0.2$ applying a single $\kappa$ value for both Aitken and accumulation modes; an even better representation of the increase in $N_d$ with $w$ was obtained when moderately hygroscopic accumulation mode particles ($\kappa_{acc}$ = 0.2) and highly hygroscopic Aitken mode particles ($\kappa_{Ait}$ = 0.8) were assumed.

While we could not further constrain the hygroscopicities of the two modes based on the available data, our results suggest that knowledge of Aitken mode particle properties is required to predict cloud droplet number concentrations in convective clouds and/or clean air masses. We conclude that in the case of a bi-modal aerosol size distribution with distinct Aitken and accumulation modes as encountered during flight AC19, $N_{d,p}$ may be significantly underestimated as compared to $N_{d,p}$ for $w \gtrsim 0.5 \, \text{m s}^{-1}$ if a single $\kappa$ for both modes is assumed that might be only appropriate for the larger accumulation mode particles. Our results also suggest that the ratio of the number concentrations of the Aitken and accumulation modes and their $\kappa$ values can influence cloud properties near cloud base differently than for one-modal aerosol size distributions. More detailed sensitivity studies of cloud properties to Aitken mode aerosol properties ($N_a$, $\kappa$) have been recently performed for wider parameter ranges to identify conditions, under which they might affect aerosol-cloud interactions (Pöhlker et al., 2021).

Our droplet closure study represents a complementary approach to constrain CCN hygroscopicity, in addition to previous studies in the same region, in which a similar $\kappa$ range (0.1–0.35) was determined for aerosol in the Amazon Basin, and a range of $0.1<\kappa<0.9$ above the ocean, based on CCN measurements and detailed analysis of chemical composition (Wex

et al., 2016; Thalman et al., 2017; Pöhlker et al., 2016, 2018). Our comparison between predicted and measured $N_d$ showed largest discrepancies at high updraft velocities ($w > 2.5$ m s$^{-1}$), which could be possibly explained by non-adiabaticity and/or entrainment of aerosol particles near cloud bases of convective clouds.

While in previous cloud droplet number closure studies the updraft velocity was often assumed to be a major factor of uncertainty, this parameter was well constrained in the current study. Implying that higher $N_d$ are formed in regions of higher updraft velocities, we sorted observed data of $N_d$ and $w$ by their frequency of occurrence ('probability matching method'). Using this approach, we reduced the uncertainty of $w$ for the $N_d$ closure. Therefore, we could largely limit our sensitivity analysis to the investigation of the importance of particle hygroscopicity and number concentration for cloud droplet number
concentrations.

    Variability in $N_a$ measurements ($\sim \pm 20\%$) translate into similar differences in predicted droplet number concentration as uncertainties assuming different $\kappa$ values, in particular at low $w$. In previous cloud droplet number closure studies, composition effects, such as slow dissolution of soluble compounds (Asa-Awuku and Nenes, 2007), reduced surface tension or variation of the water mass accommodation coefficient (Conant et al., 2004) have been inferred to explain observed droplet number
concentrations. Our analysis shows that measurement uncertainties in basic aerosol properties might equally explain such differences. If particles exceed a hygroscopicity threshold ($\kappa \gtrsim 0.3$), predicted cloud droplet number concentration becomes very insensitive to $\kappa$ when a large fraction of all particles are activated ('aerosol-limited regime'). In the presence of a distinct Aitken mode, the parameter space ($w$, $N_a$, $\kappa$) at which this regime prevails is suggested to be shifted to even higher updraft velocity regimes than in the presence of monomodal accumulation mode size distributions.

*Data availability.*  The data used in this study can be found at https://halo-db.pa.op.dlr.de/mission/5.

*Author contributions.*  RCB, BE, MLP led the analyses and the manuscript preparation. The measurements of aerosol and cloud properties were conducted and analyzed by RCB, DF, BAH, TJ, OOK, CP, DS, CV, AW, MLP. The measurements were led by MOA, MW, UP. The modeling studies were performed and interpreted by RCB, BE, DR, JDF, OL, LHP, LATM, MLP.

*Competing interests.*  The authors declare that they have no conflict of interest.

*Acknowledgements.*  The ACRIDICON–CHUVA campaign was supported by the Max Planck Society (MPG), the German Science Foundation (DFG Priority Program SPP 1294), the German Aerospace Center (DLR), and a wide range of other institutional partners. BE was supported by the French National Research Agency (ANR) (grant no. ANR-17-MPGA- 0013).

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
