# Peer review of "Cloud droplet number closure in tropical convective clouds during the ACRIDICON-CHUVA campaign"

_Atmospheric Chemistry and Physics, 2021_

## Author Comment (AC1)

**Author response to referee comments**

Referee 1

We thank the referee for the detailed comments and the good suggestions for improving the paper. We have addressed all comments as listed below which significantly improved our manuscript. Referee comments are in black, our responses in blue and manuscript text in *italic* and new text in *red*.

General changes in the manuscript:

1. We clarified how we constrain the measurements on board the HALO aircraft to investigate the relationship between w, $N_a$, $N_d$ and $\kappa$ using airborne data and model simulations. This is the first time that such measurements are performed with the proposed methodology ('Probability Matching Method' - PMM). The results from PMM analysis have shown agreement with previous studies and represents a complementary approach in which w, $N_a$, $N_d$ can be used to constrain CCN hygroscopicity;

2. We revised the abstract and conclusion section to more clearly highlight the new findings and approaches in the current study.

3. More details on the measurements of aerosol and droplet concentrations below and at cloud bases of growing convective cumuli on board the HALO aircraft are are given in terms of time, location and uncertainties (Section 2 and supplemental information);

4. We added statistical parameters to quantify the agreement in the droplet closure.

5. We added a new *Section 4.3: Sensitivities of $N_d$ predictions to w, $N_a$ and $\kappa$ where we discuss sensitivities of cloud droplet number concentration to $\kappa$, $N_a$ and w ($\xi(\kappa)$, $\xi(N_a)$ and $\xi(w)$) in the context of previous sensitivity studies.*

Below you find our specific responses to the referee comments.

**1 Referee Comment**: A number of questions remain about the measurements used in the study. Why are the CDP and CAS droplet measurements so different? What measurements were actually used during the flight? Was the entire flight averaged? Were cloud edges excluded? Were the aerosol measurements collected relatively close to the cloud droplet measurements?

Author response: We substantially extended Section 2 (Aircraft measurements) and added specific sections on aerosol and cloud measurements

[revised manuscript text omitted]

In line XX, we modified the text as follows (l. 152):

*Only cloud passes with positive w(i.e. updrafts) were considered in the analysis. Furthermore, we take into account only data of non-precipitating clouds, typically from cumulus humilis and cumulus mediocris clouds. Braga et al., 2017, have shown that*  *the PMM can be used to find the best agreement between measured and estimated $N_d$ at cloud base as a function of w.*

In addition, we included two tables in the supplement (Tables S5 and S6 in the revised manuscript)

**2 Referee comment:** While the manuscript presents data from a valuable dataset, the analysis is not thorough, substantial amounts of detailed about the case study are missing and the conclusions are weak, and not evidence based. Based on my assessment, I suggest rejecting the manuscript.

Author response: In the revised manuscript version, we made clearer how we constrain the measurements on board the HALO aircraft to investigate the relationship between w, $N_a$, $N_d$ and $\kappa$ using airborne and model simulations. Please see our response to the previous comment and our responses to the specific comments below and updates throughout the text and supplement.

**Table 1.** (Table S5 in revised manuscript) Cloud probe size intervals and central bin diameters during HALO flights.

| Cloud Probe | Size interval | Number of bins | Central bin diameter ($\mu$m) |
|---|---|---|---|
| CCP-CDP | 3-50 $\mu$m | 14 | 3.8, 6.1, 8.7, 10.9, 13.5, 17.1, 19.7, 22.5, 25.9, 28.3, 31.7, 36.6, 40.7, 44.2 |
| CAS-DPOL | 3-50 $\mu$m | 10 | 3.9, 6, 10.8 ,17.3, 22.3, 27.4, 32.4, 37.4, 42.4, 47.4 |

**Table 2.** (Table S6 in revised manuscript) Cloud base thermodynamic parameters and classification of each flight.

| Flight | Classification | Altitude [m asl.] | Air Temperature [$^{\circ}$C] | Time frame of measurements (UTC)] |
|---|---|---|---|---|
| AC07 | Arc of Deforestation | 1920 | 15 | 18:02:13 to 18:14:13 |
| AC09 | Remote Amazon | 1200 | 19.5 | 15:40:41 to 15:58:35 |
| AC18 | Remote Amazon | 1700 | 17 | 16:45:29 to 16:55:46 |
| AC19 | Atlantic Ocean | 605 | 22 | 17:28:04 to 17:37:39 |

**Specific comments:**

**3 Refereee comment:** Line 69 - please define AC. It would be more useful for the reader to name the cases by their attributes than their flight number. Something like LPC- low particle concentration, MCP1,MCP2 –moderate particle concentration...etc. Author response: The flights are label with AC and a running number, we did make this clear in the manuscript. We prefer to continue using these labels in our study for consistency with previous studies on the ACRIDICON campaign in which the same flight names were used, e.g., (Braga et al., 2017; Cecchini et al., 2017; Holanda et al., 2020; Wendisch et al., 2016). However, we added a description of the condition to the flight number.
We added to the manuscript (l. 67):
*Figure 1a shows the measurement region for the flights analyzed in this study (the flights are labelled with 'AC' and a running number, in agreement with the naming e.g., (Wendisch et al., 2016). We added a new Table S6, that also provides a characterization of the air masses)*

**4 Refereee comment:** Line 71- What do you mean by environment air? subsaturated air?
Author response: We revised the text as follows (l. 136):
*This categorization was applied to avoid cloud passes well mixed with subsaturated environment air (RH < 100%) and counts of haze particles, typically found at cloud edges.*

**5 Refereee comment:** Line 73- By "at cloud base" do you actually mean at cloud base or was it slightly above or below

cloud base?

Author response: We assume that measurements was taken slightly above cloud base. To investigate at which height above cloud base measurements was performed the measured liquid water content (LWC) was compared to the modeled LWC and the best match was found at 20 m above cloud base as shown in Figure 2. We made this more clear by adding the following sentence into the manuscript (l. 128):

*We refer in the current study to the measurements closest to cloud base as 'cloud base' measurements, even if the actual cloud base might have been slightly below this altitude of measurements (Section 3.2.2 and Figure 2).*

**6 Referee comment:** Line 78 – where were these size distribution measurements made relative to the cloud measurments? Were they averaged before fitted to log normal distributions?

Author response: the measurement strategy was developed such that measurements were made within at most 10 minutes and 60 km from each other, respectively. This was performed to assure that droplet measurements at cloud base pertain to the same air mass as the aerosol measurements below cloud base. We made this more clear by adding a map showing the flight segment below and at cloud base in Figure 1 and adding the following statement to the manuscript (l. 69):

*The measurement strategy was developed such that measurements were made within at most 10 minutes and 60 km from each other. This was performed to assure that droplet measurements at cloud base pertain to the same air mass as the aerosol measurements below cloud base.*

The aerosol size distributions was averaged first and after this the fitting was performed. This is well described in the revised Methodology section (l. 98):

*The geometric mean of the aerosol size distribution and $N_{CN}$ below cloud were calculated. The mean aerosol size distribution was fitted by one modal lognormal distributions.*

**7 Referee comment:** Line 80 – Please provide more detail on how the Aitken mode was inferred. It is unclear since the UHSAS only measured as low as 60 nm.

Author response: We added the following clarification to the text (l. 98ff).

*The geometric mean of the aerosol size distribution and $N_{CN}$ below cloud were calculated. The mean aerosol size distribution was fitted by one modal lognormal distributions. The integral of the fit for the aerosol size distribution should be similar to $N_{CN}$ if mainly accumulation mode particles are present. This was fulfilled for AC07, AC09 and  AC18, but not for AC19 (Tables S1-S4).  For this latter flight, the integrated number concentration of the monomodal lognormal fit made up approximately half of the total $N_{CN}$. This discrepancy led to the assumption that a significant number concentration of particles in the size range of Aitken mode particles were present during AC19, but not captured by the UHSAS measurements. Consequently, a bimodal ASD shape was inferred. The geometric parameters for the lognormal distribution assumed for measurements during Flight AC19 were based on averages of bimodal aerosol size distributions measured above the ocean in previous studies (Figure S4) (Wex et al., 2016; Quinn et al., 2017; Gong et al., 2019). The resulting shape of the two modes based on literature data was weighted by the difference between UHSAS and CPC measurements (Table S4). The number concentrations of all fitted aerosol size distributions were normalized to the measured $N_{CN}$. The variability of the*

*aerosol number size distributions was calculated by the standard deviation on average $\sim 10\%$ and up to $\sim 20\%$ for very clean conditions. As a conservative approach $\sim 20\%$ was used in our model sensitivity study to take into account the impact of this variability on cloud droplet number concentration (Section 4.2). All concentrations are reported for normalized atmospheric conditions (Corrected for standard conditions (STP): T = 273.15°C and p = 1013.25 mbar).*

**8 Referee comment:** Line 91 – did you demean the updraft velocity? What do you mean by passes with only positive w were considered? Surely there were often some Negative values? I am guessing you mean you just excluded negative values?

Author response: We added the following information in the manuscript.

line 142: *The thermal instability in the boundary layer promotes the formation of clouds consisting of regions with updrafts and downdrafts. At cloud bases, the variability in vertical velocities and droplet concentration is high due to air turbulence. Clouds develop in updrafts, and during their vertical development the continued movement as a turbulent eddy adds a large random component to the relationship of w with $N_{d,m.}$*

line 145. *These intrinsic characteristics of clouds reduce the confidence that a measured w in the cloud led to the simultaneously measured $N_{d,m}$. Such inconsistencies often result in poor correlations of w and $N_{d,m}$.*

line 152 *Only cloud passes with positive w (i.e. updrafts) were considered in the analysis. Furthermore, we take into account only data of non-precipitating clouds, typically from cumulus humilis and cumulus mediocris clouds.*

**9 Referee comment:** Line 106 – Was there any drizzle in the measured clouds? It makes sense to exclude collision coalescence in a parcel model since it cannot be parameterized will with a 0D model, however can you confirm with measurements and results from previous modeling studies that Collision/coalescence are negligible in the clouds you studied (at least the lower 70 m).

Author response: No, we did not have drizzle or rain at cloud base; only non-precipitating clouds were considered (cf our response to previous comment). This was checked by CIPgs. We added also more information regarding previous model studies of collision/coalescence (l. 168)

*Collision/coalescence processes are not considered as we restrict our analysis to heights near cloud base where droplets are relatively small and the cloud droplet size distribution is narrow. Under such conditions, collision-coalescence is likely negligible (Rosenfeld, 2018; Shaw et al., 1998; Xue et al., 2008).*

**10 Referee comment:** Line 112 - "In order to determine the height at which $N_{d,m}$ and $N_{d,p}$ should be compared, simulations were performed using the measured aerosol particle size distributions and an assumed hygroscopicity of k= 0.1, together with w measured at cloud base." It is not clear how this I used to determine the height at which $N_{d,m}$ and $N_{d,p}$ should be compared.

Author response: We have improved the text (l. 176ff):

*In order to determine the height at which $N_{d,m}$ and $N_{d,p}$ should be compared, the measured liqued water contend (LWC) was compared to the simulated LWC using the aerosol size distribution for the different flights together with w measured at cloud base and assumed hygroscopicity of $\kappa = 0.1$.*

*and an assumed hygroscopicity of κ = 0.1, together with w measured at cloud base.Under adiabatic conditions, $N_{d,p}$ is predicted to be approximately constant at ∼ 20 m above the level of the maximum supersaturation $S_{max}$ (Fig. S6). Figure 2 shows the frequency of measured LWC and the modeled LWC at different heights. At ∼ 20 m above cloud base the LWC measured with the highest frequency and the modeled LWC is the same. For this reasoned the model results at 20 m above cloud base are compared to the measured cloud droplet number concentrations in the scope of this study.*

**11 Referee comment:** Line 114- make it clear that this is a result from your adiabatic parcel model, not measurements.

Author response: We revised the sentence as follows (l. 178):

*Under adiabatic conditions, $N_{d,p}$ is predicted to be approximately constant at  20 m above the level of the maximum supersaturation Smax (Fig.S3).*

**12 Referee comment:** Line 120 "in the following..." Section?

Author response: Yes, we added 'Section' (now: Line 178).

**13 Referee comment:** Line 133 – how do you define best agreement? Absolute concentration? Percentage difference?

Author response: The simulated values for droplet concentration that have the best agreement with measurements as a function of κ and w are defined based on smaller bias, root mean square and standard error. We added the following tables in the supplement showing these statistics (absolute bias, RMSE, mean absolute error and ratio, new Tables S7 – S18 in the supplement).

**14 Referee comment:** Line 145 – For lateral entrainment to increase droplet concentration, the entrained concentration would need to be both CCN active and higher than the below cloud concentration. I have never seen a case where this has occurred. Can you cite a relevant source? Also, even if there was lateral entrainment, at altitudes so close to cloud base, it is unlikely that there is a significant influence from lateral entrainment. Entrainment would also likely dry the air, decreasing the supersaturation, further decreasing the chance of increasing the droplet concentration.

Author response: We agree with the referee that it seems unlikely that entrainment leads frequently to increased particle and therefore droplet concentrations. We weakened this statement and also removed at other places any text about entrainment (l. 224)

*Depending on the conditions, entrainment has been shown to lead also to the opposite trend, i. e., to the decrease of $N_d$ (Calmer et al., 2019). However, while entrainment of biomass burning aerosol may be possible, we do not have any quantitative information on such processes.*

**15 Referee comment:** Line 147 – Entrainment typically leads to a decrease in $N_d$.

Author response: We agree with the referee; please see our response to Comment 14.

**16 Referee comment:** Line 150 – why is that unlikely?

Author response: This is now explained in more detail in the revised manuscript (l. 226):

*While also particles of different hygroscopicities and activation thresholds depending on w might explain the trends in Fig.*

*3a-c, there is no indication of higher hygroscopicity of smaller accumulation mode aerosol particles during the Amazonian dry season (e.g., Pöhlker et al., 2016, 2018). In air masses of different origin, aerosol particles would likely not only exhibit different chemical composition and hygroscopicity but also large variability in their particle number concentrations. Given the relatively small standard deviations in the measured $N_a$ (Tables S1 - S4), we are confident that the sampled aerosol populations did not have large variability in their composition.*

We have extended Tables S1-S4 with additional information. We also report now all concentrations for normalized atmospheric conditions (STP corrected to T = 273.15°C and p = 1013.25 mbar).

**Tables S1 - S4**: Geometric mean and standard deviation of atmospheric parameters measured below cloud base. Note that the mean diameter is a fit parameter of the average aerosol size distribution and the error is 15 % according to (Cai et al., 2008; Moore et al., 2021).

Flight AC07
6-09-2014 17:53:00 to 17:55:25 UTC

| Parameter | Mean $\pm$ SD |
|---|---|
| Altitude [m asl.] | $1800 \pm 0.05$ |
| Air Temperature [°C] | $18 \pm 0.15$ |
| Pressure [hPa] | $820 \pm 0.25$ |
| Relative Humidity [%] | $95 \pm 3$ |
| $N_{CN}$ [cm$^{-3}$] | $2417 \pm 42$ |
| $N_{UHSAS}$ [cm$^{-3}$] | $2024 \pm 162$ |
| $d_{acc}$ [nm] | $147 \pm 22$ |

Flight AC09
11-09-2014 15:31:31 to 15:38:20 UTC

| Parameter | Mean $\pm$ SD |
|---|---|
| Altitude [m asl.] | $933 \pm 0.05$ |
| Air Temperature [°C] | $22.5 \pm 0.15$ |
| Pressure [hPa] | $938 \pm 0.25$ |
| Relative Humidity [%] | $87 \pm 3$ |
| $N_{CN}$ [cm$^{-3}$] | $737 \pm 58$ |
| $N_{UHSAS}$ [cm$^{-3}$] | $686 \pm 59$ |
| $d_{acc}$ [nm] | $140 \pm 22$ |

Flight AC18
28-09-2014 16:39:00 to 16:43:59 UTC

| Parameter | Mean $\pm$ SD |
|---|---|
| Altitude [m asl.] | $1286 \pm 0.05$ |
| Air Temperature [°C] | $20.7 \pm 0.15$ |
| Pressure [hPa] | $876 \pm 0.25$ |
| Relative Humidity [%] | $81.8 \pm 2$ |
| $N_{CN}$ [cm$^{-3}$] | $809 \pm 20$ |
| $N_{UHSAS}$ [cm$^{-3}$] | $707 \pm 83$ |
| $d_{acc}$ [nm] | $140 \pm 21$ |

Flight AC19
30-09-2014 17:23:38 to 17:27:31 UTC

| Parameter | Mean $\pm$ SD |
|---|---|
| Altitude [m asl.] | $452 \pm 0.05$ |
| Air Temperature [°C] | $23.5 \pm 0.15$ |
| Pressure [hPa] | $960 \pm 0.25$ |
| Relative Humidity [%] | $93.7 \pm 3$ |
| $N_{CN}$ [cm$^{-3}$] | $428 \pm 138$ |
| $N_{UHSAS}$ [cm$^{-3}$] | $227 \pm 52$ |
| $N_{ait}$ [cm$^{-3}$] | $\sim 201$ |
| $N_{acc}$ [cm$^{-3}$] | $227 \pm 52$ |
| $d_{ait}$ [nm] | $\sim 37$ |
| $d_{acc}$ [nm] | $136 \pm 20$ |

**17 Referee comment:** Line 155- What evidence is there for this cloud being impacted by marine air? Having a bimodal distribution does not make an aerosol distribution impacted by marine air. It is very likely that all of the particle distributions

have a bimodal distribution, however you cannot tell because you are limited to a minimum particle size measurement of 60 nm by the UHSAS. It is still unclear how you obtained an Aitken mode for AC19.

Author response: We agree with the referee that bimodality does not necessarily imply that the air mass had marine origin and vice versa. As it was misleading in the text, we changed it as follows (l. 238):

*The air masses below cloud encountered during flight AC19 were mostly impacted by marine air*  *(as supported by prior back trajectory analysis (Section S1 and Holanda et al. (2020))) and exhibited a bi-modal aerosol size distribution with low $N_{d,m}$ (Figure 1c).*

We had strong indications that indeed a monomodal aerosol distribution was not sufficient to explain the observed $N_a$. We added more details on the fitting procedure of the bimodal size distributions (see our response to Referee Comment 7).

**18 Referee comment:** Line 157 – this result suggests that the aerosol you measured and the aerosol that entered the cloud are not from the same population.

Author response: We added more details on the aerosol and cloud measurements in Section 2. (See our responses to your general comments 1 and 2 at the beginning of this response). In addition, we also added more explanation on the reasons why the $N_d$ closure is worse for AC19 than for the other flights. In brief, we argue that likely the assumption of identical $\kappa$ values for both modes is an oversimplification and that properties for Aitken and accumulation modes need to be taken into account.

**19 Referee comment:** Line 165 The following two quotes from your text are inconsistent with your argument "The chemical composition of Aitken mode particles often differs significantly from that of accumulation mode particles, which are more aged and internally mixed" "Marine particles often show similar hygroscopicity in both Aitken and accumulation modes"

Author response: We agree with the referee that the text was contradictory as it was written. We removed the respective sentence:

**20 Referee comment:** Line 168 – You have indicated it is possible that the hygroscopicity of these particles may be inconsistent with the values you used because you think they are marine aerosol. You have the ability to test this hypothesis with your parcel model, but choose not to. Why?

Author response: We thank referee for this suggestion and agree that further sensitivity simulations added value to our study. We performed additional studies using all combinations of $\kappa$ = 0.1 and 0.6 (and 0.8/0.1) for the two modes. Results are shown in Figure 3 e and discussed as follows (l. 253ff):

*To account for different hygroscopicities in Aitken and accumulation modes, we performed further sensitivity analyses using combinations of $\kappa$ = 0.1 and 0.6 for the two modes (Figure R1-2e). It is obvious that the choice of $\kappa$ for the Aitken mode ($\kappa_{Ait}$) does not affect $N_{d,p}$ for $w \leq\sim 1$ m s$^{-1}$ in the presence of very hygroscopic accumulation mode particles ($\kappa_{acc}$ = 0.6) or below $w \leq\sim 0.5$ m s$^{-1}$ with $\kappa_{acc}$ = 0.1, respectively. Even assuming rather extreme values of $\kappa_{Ait}$ = 0.8 cannot fully reproduce the large increase in $N_d$ at $w \geq\sim 1.5$ m s$^{-1}$ as observed by the CAS probes; assuming very hygroscopic Aitken mode and less hygroscopic accumulation mode particles can approximately reproduce the trend in $N_{d,m}$ from the CDP.*

*Varying $\kappa_{acc}$ from 0.1 to 0.6 leads to a large increase of $N_{d,p}$ at all w. The corresponding change in $N_{d,p}$ by increasing $\kappa_{Ait}$ is much smaller. The reason for this relatively smaller sensitivity of $N_{d,p}$ to $\kappa_{Ait}$ is the fact that the supersaturation in the cloud is mostly controlled by the droplet growth on accumulation mode particles. The sensitivity of $N_{d,p}$ formed on Aitken mode particles to $\kappa_{acc}$ is slightly larger if $\kappa_{acc} = 0.1$ as compared to $\kappa_{acc} = 0.6$ because in the latter case, the supersaturation is efficiently suppressed preventing a higher number of Aitken mode particles from activating. Overall we can conclude that assuming different $\kappa$ values for accumulation and Aitken mode leads to a better representation of the observed trends of $N_{d,m}$ with w (Tables S16 ad S17). However, in the absence of more information on the particle hygroscopicity we cannot state with certainty that indeed the assumptions of the two $\kappa$ values are appropriate for this aerosol population. Figure 3d clearly shows that the simplified assumption of a single $\kappa$ is not appropriate to infer $N_{d,p}$ for low aerosol loading and when the particle number concentrations of the accumulation and Aitken modes are comparable. By using a single $\kappa$ value, we cannot reproduce the observed continuously strong increase of $N_{d,m}$ for the whole w range. Instead we predict a smaller increase at $w \sim 1\ m\ s^{-1}$, i.e., a flattening of the curve.*

Figure 3 was accordingly updated. Please see Figure R1-2. As indeed the simulations assuming $\kappa_{Ait} = 0.6$ and $\kappa_{acc} = 0.1$ or $\kappa_{Ait} = 0.8$ and $\kappa_{acc} = 0.2$ give much better results, we also discuss them now in the text of the abstract and summary and conclusion section:

**Abstract**: *Above the ocean, fair agreement was obtained assuming an average hygroscopicity of $\kappa \sim 0.2$ (deviations $\leq \sim 16\%$) and further improvement was achieved assuming different hygroscopicities for Aitken and accumulation mode particles ($\kappa_{Ait} = 0.8$, $\kappa_{acc} = 0.2$; deviations $\leq \sim 10\%$), which may reflect secondary marine sulfate particles. Our results indicate that Aitken mode particles and their hygroscopicity can be important for droplet formation at low pollution levels and high updraft velocities in tropical convective clouds.*

**Summary and conclusions**: *Above the western Atlantic best $N_d$ closure was achieved for $\kappa \sim 0.2$ applying a single $\kappa$ value for both Aitken and accumulation modes; an even better representation of the increase in $N_d$ with w was obtained when moderately hygroscopic accumulation mode particles ($\kappa_{acc} = 0.2$) and highly hygroscopic Aitken mode particles ($\kappa_{Ait} = 0.8$) were assumed.*

**21 Referee comment:** Line 172- you should include sensitivity calculations.

Author response: We added the new Figure S7 to the supplement. In addition, we added some detail and discussion about the calculation and interpretation of the sensitivities in the new Section 4.3 (Please our detailed response to Referee Comment 28.)

**22 Referee comment:** Line 175- do you mean additional aerosol? Additional activation would surely lead to additional activated particles.

Author response: We agree with the referee that the sentence was misleading. We reworded it as follows (l. 407):

*In these studies, it was demonstrated that  the sensitivity of $N_d$ to becomes small.  Our analysis shows that measurement uncertainties in basic aerosol properties might equally explain such differences. If particles exceed a hygroscopicity threshold ($\kappa > \sim 0.3$), predicted cloud droplet number concentration becomes very insensitive to $\kappa$ when a large fraction of all particles are activated ('aerosol-limited regime').*

[Figure]

**Figure R1-2.** (= Figure 3 in the revised manuscript) Cloud droplet number concentration ($N_d$) as a function of updraft velocity near cloud base of convective clouds during flights: a) AC07, b) AC09, c) AC18, d) and e) AC19. The measured updraft velocities are based on the "probability matching method" (PMM) using the same percentiles for updraft velocity and $N_{d,m}$ (Section 3.1). The black diamond and triangle symbols represent $N_{d,m}$ near cloud base with the CAS-DPOL and CCP-CDP probes, respectively. Measurement uncertainties (indicated by error bars) are $\sim 21\%$ and $\sim 10\%$ for CAS-DPOL and CCP-CDP data (Braga et al. (2017)). The colored lines in panels a)-d) show $N_{d,p}$ assuming a single $\kappa$ value for both modes (labeled on the left). Panel e) shows $N_{d,p}$ based on simulations assuming different values of $\kappa$ for Aitken and accumulation mode particles.

[Figure]

**Figure R1-3.** (= Figure S7 in the revised manuscript) Modeled sensitivity of droplet number concentration ($N_d$) to changes in the hygroscopicity parameter $\kappa$, vertical velocity ($w$) and aerosol number concentration ($N_c$) for the measured conditions during flight AC07 (arc of deforestation), AC09 and AC18 (remote Amazon) and AC19 (Atlantic ocean).

**23 Referee comment:** Line 178 – "high sensitivities of $N_d$ to the chemical composition of Aitken mode particles might affect cloud properties" This statement is redundant.

Author response: We reworded the sentence as follows (l. 277):

 Anttila and Kerminen (2007) showed in a model study focusing only on Aitken mode particles that $N_d$ is highly sensitive to the chemical composition of Aitken mode particles.

**24 Referee comment:** Line 179 – comparable in composition? Concentration? This is unclear. This statement about kappa contradicts your statement in line 167.

Author response: We are not sure which line the referee is referring to. As the 'comparable' in the following sentence may have led to the confusion, we clarified it as follows (l. 268):

when  the particle number concentrations of the accumulation and Aitken modes are comparable.

With the regard to the statement about $\kappa$ in the two modes, we added some findings from our most recent study (Pöhlker et al., 2021) that was published in ACPD nearly concurrently with the ACPD version of the present manuscript. There we show that the sensitivities of $\kappa$ and $N_a$ to $N_d$ are different for accumulation and Aitken mode particles, respectively.

Therefore, our statements are not contradicting each other as in the previous text, we referred to accumulation mode particles. We clarify this now as follows (l. 278):

*In our recent model study, we systematically explored the extent to which the presence of an Aitken mode might significantly affect $N_d$ as a function of updraft velocity (Pöhlker et al., 2021). In that study, we show that the sensitivities of $N_{d,p}$ are different to the properties ($N_a$, $\kappa$) of accumulation and Aitken mode particles, respectively. Generally, we find that $N_{d,m}$ is not highly sensitive to Aitken mode particle properties in the presence of a dominant accumulation mode, which is in agreement to our results in Figures 3 and S7.*

**25 Referee comment:** Line 186- how was this 30% uncertainty calculated?

The uncertainty is actually only 20%, being the sum of 10% uncertainty for aerosol size distribution measurements and 10% uncertainty in the measurement $N_d$. We noticed that the 30% mentioned in the previous manuscript version were too high as one of the uncertainty was erroneously double-counted.

We added the error bars for $N_{d,m}$ in the figures. To explore the sensitivities of $N_{d,p}$ to an even wider range of $N_a$, we included now model results for different $N_a$ ($\pm 20\%$, $\pm 30\%$, $\pm 40\%$) to not only cover the uncertainty and variability in measurements but also show the sensitivity of $N_d$ to $N_a$. In addition we expend the discussion about uncertainty and variability of the measured aerosol size distribution. Accordingly, the following changes have been bade in the manuscript (l. 92ff):

*The total particle number concentration in the size range of $\sim 10$ nm to $\sim 500$ nm ($N_{CN}$) below cloud base were measured using the Aerosol Measurement System (AMETYST), the uncertainty of these measurements is estimated to be $10\%$ (Andreae et al., 2018). $N_{CN}$ was measured by a butanol-based condensation particle counter (CPCs, modified Grimm CPC 5.410 by Grimm Aerosol Technik, Ainring, Germany) with a flow of 0.6 L min$^{-1}$. Particle losses in the sampling lines have been estimated and taken into account with the particle loss calculator by von der Weiden et al. (2009). Typical uncertainties of CPC measurements are on the order of $\sim 10\%$ (Petzold et al., 2011).*

*The geometric mean of the aerosol size distribution and $N_{CN}$ below cloud were calculated. The mean aerosol size distribution was fitted by one modal lognormal distributions. The integral of the fit for the aerosol size distribution should be similar to $N_{CN}$ if mainly accumulation mode particles are present. This was fulfilled for AC07, AC09 and AC18, but not for AC19 (Tables S1-S4). For this latter flight, the integrated number concentration of the monomodal lognormal fit made up approximately half of the total $N_{CN}$. This discrepancy led to the assumption that a significant number concentration of particles in the size range of Aitken mode particles were present during AC19, but not captured by the UHSAS measurements. Consequently, a bimodal ASD shape was inferred. The geometric parameters for the lognormal distribution assumed for measurements during Flight AC19 were based on averages of bimodal aerosol size distributions measured above the ocean in previous studies (Figure S4) (Wex et al., 2016; Quinn et al., 2017; Gong et al., 2019). The resulting shape of the two modes based on literature data was weighted by the difference between UHSAS and CPC measurements (Table S4). The number concentrations of all fitted aerosol size distributions were normalized to the measured $N_{CN}$. The variability of the aerosol number size distributions was calculated*

*by the standard deviation on average ∼ 10 % and up to ∼ 20 % for very clean conditions. As a conservative approach ∼ 20 % was used in our model sensitivity study to take into account the impact of this variability on cloud droplet number concentration (Section 4.2). All concentrations are reported for normalized atmospheric conditions (Corrected for standard conditions (STP): T = 273.15°C and p = 1013.25 mbar).*

Figure 4 was replaced by the following figure and the text was adjusted accordingly.

[Figure]

**Figure R1-4.** (= Figure 4 in the revised manuscript) Cloud droplet number concentration ($N_d$) as a function of updraft velocity near cloud base of convective clouds during flights: a) AC07, b) AC09, c) AC18, d) and e) AC19. The measured updraft velocities are based on the "probability matching method" (PMM) using the same percentiles for updraft velocity and $N_{d,m}$ (Section 3.1). The black diamond and triangle symbols represent $N_{d,m}$ near cloud base with the CAS-DPOL and CCP-CDP probes, respectively. Measurement uncertainties (indicated by error bars) are ∼ 21% and ∼ 10% for CAS-DPOL and CCP-CDP data (Braga et al. (2017)). The lines show $N_{d,p}$ assuming the uncertainty range of $N_a$ measurements, colored-coded by $\Delta N_a$ [%].

**26 Referee comment:** Line 215 – non-adiabatic conditions like entrainment would not increase particle concentration, only decrease.

Author response: We agree with the referee that the discussion of entrainment as an explanation for resulting higher $N_d$ does not seem likely. Therefore, we weakened this statement as follows (l.394):

 *Our comparison between predicted and measured $N_d$ showed largest discrepancies at high updraft velocities (w > 2.5 m s-1), which  could be possibly explained by non-adiabaticity and/or entrainment of aerosol particles near cloud bases of convective clouds.*

**27 Referee comment:** Line 217 calculations are necessary to back this claim and could easily be performed.

Author response: We agree with the referee that the predicted differences in $N_d$ due to variation in $\kappa$ or $N_a$ could have been more quantified. Statistical parameters in Tables S7-S18 have been added and are discussed in Section 4.1 and 4.2. The new Section 4.3 also provides more information on such sensitivities in the present and previous studies.

**28 Referee comment:** Line 218: you should be able to state your conclusions based on the evidence provided in the manuscript, not by citing others work.

Author response: We agree with the referee that the placement of these references was not ideal. We moved them now to the new Section 4.3 (Sensitivities of $N_d$ predictions to w, $N_a$ and $\kappa$ in the context of previous studies and restricted the text in the Summary and conclusions Section to conclusions based on our study.

[revised manuscript text omitted]

---

## Author Comment (AC2)

**Author response to referee comments**
**Referee 2**

This manuscript showed closure results of measured and predicted cloud droplet number concentration for variable updraft speed during ACRIDICON-CHUVA campaign of 2014 where role of updraft speed, hygroscopicity and aerosol size distribution is discussed. Better closure results are obtained when k was assumed to be 0.1, updraft velocity is low and aerosol size distribution is unimodal. CCP and CAS-DPOL are used to measure cloud droplet size distribution and UHSAS with CPC for aerosol size distribution. Updraft speed is measured using Rosemount model 858 AJ probe. Overall the results could be a valuable contribution if they are backed up by proper justification. One of the major point of concern is the lack of reasoning when there is an agreement or disagreement in the closure results. It reads more like a report lacking scientific understanding of the results. I recommend the publication only if the authors improve the discussion part and add previous relevant studies for comparison and show why their approach is better than the earlier studies.

Author response: We thank the referee for the detailed comments and the good suggestions for improving the paper. We have addressed all comments as listed below which significantly improved our manuscript. Referee comments are in black, our responses in blue and manuscript text in *italic* and new text in *red*.

General changes in the manuscript:

1. We clarified how we constrain the measurements on board the HALO aircraft to investigate the relationship between w, $N_a$, $N_d$ and $\kappa$ using airborne data and model simulations. This is the first time that such measurements are performed with the proposed methodology ('Probability Matching Method' - PMM). The results from PMM analysis have shown agreement with previous studies and represents a complementary approach in which w, $N_a$, $N_d$ can be used to constrain CCN hygroscopicity;

2. We revised the abstract and conclusion section to more clearly highlight the new findings and approaches in the current study.

3. More details on the measurements of aerosol and droplet concentrations below and at cloud bases of growing convective cumuli on board the HALO aircraft are are given in terms of time, location and uncertainties (Section 2 and supplemental information);

4. We added statistical parameters to quantify the agreement in the droplet closure.

5. We added a new *Section 4.3: Sensitivities of $N_d$ predictions to w, $N_a$ and $\kappa$ where we discuss sensitivities of cloud droplet number concentration to $\kappa$, $N_a$ and w ($\xi(\kappa)$, $\xi(N_a)$ and $\xi(w)$) in the context of previous sensitivity studies.*

Below you find our specific responses to the referee comments.

**Specific Comments:**

**1 Referee comment:** Line 66: Height of the cloud base? Author response: The cloud base heights were different for each region of study. We provide this information for each flight in a new Table in the supplement.

**2 Referee comment:** Line 63-64: Purpose of using two probes: CCP and CAS-DPOL.
Author response: This was performed to test our methodology with cloud probes that use different characteristics (such as particle inlet, sampling area of detection, sizes sensitivities etc.) to measure cloud particles. We substantially extended Sections 2 and 3 and address the specific referee comment in line 118:
*These probes have different measurement characteristics such as particle inlet, sampling area of detection, size sensitivities etc. The CCP-CDP is an open-path instrument that detects forward-scattered laser light from cloud particles as they pass through the CDP detection area (Lance et al., 2010). CAS-DPOL collects forward-scattered light to determine particle size and number that pass the sampling area centered in an inlet shaft that guides the airflow. CCP-CDP and CAS-DPOL has similar values of uncertainty ($\sim$ 10%) in the sample area. However, particle velocities in the sampling tube may be modified by the CAS tube when compared to the open path instruments (like CCP-CDP). This results in an additional uncertainty in the droplet number concentration measured by CAS-DPOL. During the ACRIDICON-CHUVA campaign the resulting uncertainty in the droplet concentration measured by CCP-CDP and CAS-DPOL were $\sim$ 10% and $\sim$ 21 %, respectively (Braga et al., 2017a).*

**3 Referee comment:** Line 84 and 186: How is the uncertainty of 30% is estimated?
The uncertainty is actually only 20%, being the sum of 10% uncertainty for aerosol size distribution measurements and 10% uncertainty in the measurement $N_d$. We noticed that the 30% mentioned in the previous manuscript version were too high as one of the uncertainty was erroneously double-counted.
We added the error bars for $N_{d,m}$ in the figures. To explore the sensitivities of $N_{d,p}$ to an even wider range of $N_a$, we included now model results for different $N_a$ ($\pm$20 %, $\pm$30 %, $\pm$40 %) to not only cover the uncertainty and variability in measurements but also show the sensitivity of $N_d$ to $N_a$. In addition we expend the discussion about uncertainty and variability of the measured aerosol size distribution. Accordingly, the following changes have been bade in the manuscript (l. 92ff):
*The total particle number concentration in the size range of $\sim$ 10 nm to $\sim$ 500 nm ($N_{CN}$) below cloud base were measured using the Aerosol Measurement System (AMETYST), the uncertainty of these measurements is estimated to be 10 % (Andreae et al., 2018). $N_{CN}$ was measured by a butanol-based condensation particle counter (CPCs, modified Grimm CPC 5.410 by Grimm Aerosol Technik, Ainring, Germany) with a flow of 0.6 L min$^{-1}$. Particle losses in the sampling lines have been estimated and taken into account with the particle loss calculator by von der Weiden et al. (2009). Typical uncertainties of CPC measurements are on the order of $\sim$10 % (Petzold et al., 2011).*
*The geometric mean of the aerosol size distribution and $N_{CN}$ below cloud were calculated. The mean aerosol size distribution was fitted by one modal lognormal distributions. The integral of the fit for the aerosol size distribution should be similar to $N_{CN}$ if mainly accumulation mode particles are present. This was fulfilled for AC07, AC09 and AC18, but not for AC19 (Tables S1-S4). For this latter flight, the integrated number concentration of the monomodal lognormal fit made up approximately half of the total $N_{CN}$. This discrepancy led to the assumption that a significant number concentration of particles in the size range*

*of Aitken mode particles were present during AC19, but not captured by the UHSAS measurements. Consequently, a bimodal ASD shape was inferred. The geometric parameters for the lognormal distribution assumed for measurements during Flight AC19 were based on averages of bimodal aerosol size distributions measured above the ocean in previous studies (Figure S4) (Wex et al., 2016; Quinn et al., 2017; Gong et al., 2019). The resulting shape of the two modes based on literature data was weighted by the difference between UHSAS and CPC measurements (Table S4). The number concentrations of all fitted aerosol size distributions were normalized to the measured $N_{CN}$. The variability of the aerosol number size distributions was calculated by the standard deviation on average $\sim$ 10 % and up to $\sim$ 20 % for very clean conditions. As a conservative approach $\sim$ 20 % was used in our model sensitivity study to take into account the impact of this variability on cloud droplet number concentration (Section 4.2). All concentrations are reported for normalized atmospheric conditions (Corrected for standard conditions (STP): T = 273.15°C and p = 1013.25 mbar).*

Figure 4 was replaced by Figure R2-1 and the text was adjusted accordingly

**4 Referee comment:** Line 103: What will be the effect of size dependent hygroscopicity is assumed for external mixing state.

Author response: Assuming external mixing states will result in similar trends of predicted $N_d$ as a function of hygroscopicity. However, previous sensitivity studies have shown that in particular for marine air masses, the assumption of internally mixed aerosol is more appropriate. We added the following text in l. 163ff:

*It is assumed that the aerosol particles are internally mixed with identical hygroscopicity ($\kappa$) of all particles. This assumption was made based on previous sensitivity studies that have shown that for marine and aged continental air masses internal mixtures are suitable approximations (Ervens et al., 2010).*

We also performed additional sensitivity studies in which we assumed different $\kappa$ values for Aitken and accumulation modes, respectively. The results of these simulations are shown in Figure 3 and discussed in the text (l. 253ff).

*To account for different hygroscopicities in Aitken and accumulation modes, we performed further sensitivity analyses using combinations of $\kappa$ = 0.1 and 0.6 for the two modes (Figure ??e). It is obvious that the choice of $\kappa$ for the Aitken mode ($\kappa_{Ait}$) does not affect $N_{d,p}$ for w $\leq\sim$ 1 m s$^{-1}$ in the presence of very hygroscopic accumulation mode particles ($\kappa_{acc}$ = 0.6) or below w $\leq\sim$ 0.5 m s$^{-1}$ with $\kappa_{acc}$ = 0.1, respectively. Even assuming rather extreme values of $\kappa_{Ait}$ = 0.8 cannot fully reproduce the large increase in $N_d$ at w $\geq\sim$ 1.5 m s$^{-1}$ as observed by the CAS probes; assuming very hygroscopic Aitken mode and less hygroscopic accumulation mode particles can approximately reproduce the trend in $N_{d,m}$ from the CDP.*

*Varying $\kappa_{acc}$ from 0.1 to 0.6 leads to a large increase of $N_{d,p}$ at all w. The corresponding change in $N_{d,p}$ by increasing $\kappa_{Ait}$ is much smaller. The reason for this relatively smaller sensitivity of $N_{d,p}$ to $\kappa_{Ait}$ is the fact that the supersaturation in the cloud is mostly controlled by the droplet growth on accumulation mode particles. The sensitivity of $N_{d,p}$ formed on Aitken mode particles to $\kappa_{acc}$ is slightly larger if $\kappa_{acc}$ = 0.1 as compared to $\kappa_{acc}$ = 0.6, because in the latter case the supersaturation is efficiently suppressed preventing a higher number of Aitken mode particles from activating. Overall we can conclude that assuming different $\kappa$ values for accumulation and Aitken mode leads to a better representation of the observed trends of $N_{d,m}$ with w (Tables S16 and S17). However, in the absence of more information on the particle hygroscopicity we cannot state with certainty that the assumptions of the two $\kappa$ values are appropriate for this aerosol population. Figure ??d clearly shows that the simplified as-*

[Figure]

**Figure R2-1.** Cloud droplet number concentration ($N_d$) as a function of updraft velocity near cloud base of convective clouds during flights: a) AC07, b) AC09, c) AC18, d) and e) AC19. The measured updraft velocities are based on the "probability matching method" (PMM) using the same percentiles for updraft velocity and $N_{d,m}$ (Section 3.1). The black diamond and triangle symbols represent $N_{d,m}$ near cloud base with the CAS-DPOL and CCP-CDP probes, respectively. Measurement uncertainties (indicated by error bars) are $\sim 21\%$ and $\sim 10\%$ for CAS-DPOL and CCP-CDP data (Braga et al. (2017a)). The lines show $N_{d,p}$ assuming the uncertainty range of $N_a$ measurements, colored-coded by $\Delta N_a$ [%].

*sumption of a single $\kappa$ is not appropriate to infer $N_{d,p}$ for low aerosol loading and when the particle number concentrations of the accumulation and Aitken modes are comparable. By using a single $\kappa$ value, we cannot reproduce the observed continuously strong increase of $N_{d,m}$ for the whole w range. Instead we predict a smaller increase at $w \sim 1\ m\ s^{-1}$, i.e., a flattening of the curve.*

**5 Referee comment:** Line 133: Why there is a deviation between measured and modelled $N_d$ at low w?

Author response: The reason for these deviations is mostly associated with the initial ASD assumed to input the model. For each flight, we have assumed the averaged ASD measured below cloud bases of convective cumuli. This means that it is expected that some disagreement between model and measurements may be found for cloud passes in which $N_d$ were formed from ASDs that the total aerosol number concentration is below or larger than 30% (aerosol concentration uncertainty). Furthermore, cloud passes within pollution plumes from biomass burning may add additional disagreement especially at higher updraft speeds (w > 2.5 m s$^{-1}$).

**6 Referee comment:** Line 129-130: What is the implication?

Author response: The fact that for all flights a single value of $\kappa$ can reproduce the measured $N_d$ within all other uncertainties is one of the main findings of our study. We point out that this $\kappa$ value is an effective value as used in many previous studies to fit the CCN activity. The implications of this findings are that the description of CCN activation and cloud droplet formation for similar air masses can be satisfactorily described by this $\kappa$ value. However, as we show in our in our additional sensitivity studies (see response to Comment 4), that in the presence of bimodal aerosol size distributions even better closure maybe reached if different $\kappa$ values for Aitken and accumulation modes are applied we added the following text to the abstract and conclusions:

**Abstract**: *Above the ocean, fair agreement was obtained assuming an average hygroscopicity of $\kappa \sim 0.2$ (deviations $\leq \sim 16\%$) and further improvement was achieved assuming different hygroscopicities for* **Aitken and accumulation** *mode particles ($\kappa_{Ait}$ = 0.8, $\kappa_{acc}$ = 0.2; deviations $\leq \sim 10\%$), which may reflect secondary marine sulfate particles. Our results indicate that Aitken mode particles and their hygroscopicity can be important for droplet formation at low pollution levels and high updraft velocities in tropical convective clouds.*

**Summary and conclusions**: *Above the western Atlantic best $N_d$ closure was achieved for $\kappa \sim 0.2$ applying a single $\kappa$ value for both Aitken and accumulation modes; an even better representation of the increase in $N_d$ with w was obtained when moderately hygroscopic accumulation mode particles ($\kappa_{acc}$ = 0.2) and highly hygroscopic Aitken mode particles ($\kappa_{Ait}$ = 0.8) were assumed.*

**6 Referee comment:** Line 105: Why collision and coalescence are not considered? Is there any measurement constraint or it is not important to consider?

Author response: Measurement were only performed in non-precipitating clouds. This was checked by CIPgs. We added also more information regarding previous model studies of collision/coalescence (l. 168)

*Collision/coalescence processes are not considered as we restrict our analysis to heights near cloud base where droplets are relatively small and the cloud droplet size distribution is narrow. Under such conditions, collision-coalescence is likely negligible (Shaw et al., 1998; Xue et al., 2008; Rosenfeld, 2018; Braga et al., 2017b).*

**7 Referee comment:** Line 147: Under which specific conditions, there will be decrease of $N_d$ due to entrainment of additional aerosols?

Author response: As Referee 1 also questioned the likelihood that entrainment may increase particle concentration and thus cloud droplet concentration, we modified the text in l. 224

 *However, while entrainment of biomass burning aerosol may be possible, we do not have any quantitative information on such processes.*

We also weakened the corresponding statement in the conclusion section (l. 394):

 *Our comparison between predicted and measured $N_d$ showed largest discrepancies at high updraft velocities (w > 2.5 m s$^{-1}$), which  could be possibly explained by non-adiabaticity and/or entrainment of aerosol particles near cloud bases of convective clouds.*

**8 Referee comment:** Line 155: Why is it assumed that bimodal size distribution is due to marine air? Are there any evidences of size dependent chemical composition?

Author response: We agree with the referee that bimodality does not necessarily imply that the air mass had marine origin and vice versa. As it was misleading in the text, we changed it as follows:

*The air masses below cloud encountered during flight AC19 were mostly impacted by marine air  and exhibited a bi-modal aerosol size distribution with low $N_{d,m}$.*

We had strong indications that indeed a monomodal aerosol distribution was not sufficient to explain the observed $N_a$. We added more details on the fitting of the bimodal size distributions (l. 98ff):

*The geometric mean of the aerosol size distribution and $N_{CN}$ below cloud were calculated. The mean aerosol size distribution was fitted by a one modal lognormal distributions. The integral of the fit for the aerosol size distribution should be similar to $N_{CN}$ if mainly accumulation mode particles are present. This was fulfilled for AC07, AC09  and  AC18 but not for AC19 (Tables S1 - S4).  For this latter flight, the integrated number concentration of the monomodal lognormal fit made up approximately half of the total $N_{CN}$. This discrepancy led to the assumption that a significant number concentration of particles in the size range of Aitken mode particles were present during AC19, but not captured by the UHSAS measurements. Consequently, a bimodal ASD shape was inferred. The geometric parameters for the lognormal distribution assumed for measurements during Flight AC19 were based on averages of bimodal aerosol size distributions measured above the ocean in previous studies (Figure S4) (Wex et al., 2016; Quinn et al., 2017; Gong et al., 2019). The resulting shape of the two modes based on literature data was weighted by the difference between UHSAS and CPC measurements (Table S4). The number concentrations of all fitted aerosol size distributions were normalized to the measured $N_{CN}$.*

**9 Referee comment:** Line 182-183: The sensitivity analysis of Aitken and accumulation mode to total $N_a$ and $N_d$ should be included in this study as this is one of the highlight of this manuscript that represents scientific advancement.

Author response: We agree with the referee that our data set showed interesting results that highlight the possible importance of the individual properties of Aitken and accumulation mode particles for cloud droplet number concentration.

In the revised manuscript, we added some findings from our recent study (Pöhlker et al., 2021) that was published in ACPD nearly concurrently with the present article, and was just accepted. There we show for example, that the sensitivities of $\kappa$ and $N_a$ to $N_d$ are different for accumulation and Aitken mode particles, respectively. This was added in line 278:

*In our recent model study, we have shown that in the transitional regime, i.e., in the parameter space between the aerosol- and updraft limited regimes, as defined by Reutter et al. (2009), $N_d$ can be equally sensitive to $\kappa$ and w (Pöhlker et al., 2021). In that study, we show that with increasing $N_a$, the sensitivities to both parameters decrease; however, the sensitivity of $N_d$ to w remains higher under such conditions than that to $\kappa$.*

**References**

[revised manuscript text omitted]

---

## Author Response (AR2)

**Author response to Referee 3's comments**

Author response: We thank the referee for the detailed comments and the good suggestions for improving the paper. We have addressed all comments as listed below which significantly improved our manuscript. Referee comments are in black, our responses in blue and manuscript text in *italic* and new text in *red*.

Below you find our specific responses to the referee comments.

Referee 3

The main value of this work lies in the information offered from measurements of aerosol and cloud properties in fairly convective clouds in an interesting location.

1) The main objective(s) of this paper are unclear: a. The title indicates closure, but reasonable closure is not possible here for two reasons. One is that there are two measurements of cloud droplet number concentrations that are separated by 20-30% before considering the uncertainty in the measurements. The second reason is that there is no independent estimate of kappa. With the information presented, the model is incapable of producing closure because it is the comparison of the model and observations that is used to estimate the best kappa. This implicitly assumes that the other observations are correct, but we know that is not the case because the two Nd measurements disagree. I am not being critical of the Nd measurements: I applaud the authors for including both. My suggestion is to take your best case, in terms of measured quantities, including information available to define kappa, and attempt closure of the model with the two measurements of Nd. Your best case might be the simplest, and the one that has the best estimate of updraft speed or w; more discussion of w later. The objective of this best case would be to be able to say something about how the two measurements relate to the modelled Nd.

Author response:

(a) The focus of the paper is to describe the comparison of measured and modeled droplet concentrations at cloud bases of convective clouds. We describe a new approach using a different set of in situ measurements as it has been done before and a parcel model.

(b) We respectfully disagree that the term 'closure' is not appropriate here. We refer to the definition of 'closure', as described in previous studies, e.g. by Quinn and Coffman (1998): *"In a closure experiment, an aerosol property is measured by one or more methods and calculated from a model that is based on other independently measured properties. A comparison of the measured and calculated values can reveal inadequacies in either the measurements or the model."* Even the aforementioned study did not focus on cloud droplet number concentration, generally this definition is very well in agreement with our approach.

(c) The closure analysis was performed separately for each cloud probe in order to verify the methodology using two types of instruments. The biases of the results from both probes were expected since they have differences in measurements characteristics and uncertainties. Furthermore, as shown at Braga et al. (2017) a perfect agreement is not expected between them since they were mounted in different wings and measured different air samples. Braga et al. (2017) have shown that both probes agrees within their measurement uncertainties. This is an inherent characteristic of in situ measurements. Our study reveals the sensitivity of $N_d$ to various parameters ($\kappa$, $N_a$, $w$). To our knowledge, there are not

many studies in the literature that performed comparable studies for a such a rich data set and therefore could be used to reveal sensitivities. The statistical results that are well described for both probes within the main text and supplement.

(d) We realized that some of your text was misleading as we do not fit our results to $\kappa$ to get a 'best estimate' but rather describe our results with the adiabatic parcel model as a function of prescribed $\kappa$ values that are in the range of previously determined ones. The novelty of our analysis is the possibility to assess the relative uncertainties of various parameters based on a unique combination of measurements.

We make these points clearer throughout the revised manuscript.

b. In the abstract and later in the text, there is considerable discussion of the best kappa value. I don't see this as a major objective of this paper. There are better ways to estimate kappa, which the authors refer to, than to use such a convoluted approach that includes uncertainties. Use estimates of kappa from previous studies in the region.

The focus of the paper is to describe the closure analysis at cloud bases of convective clouds. The referee is right that indeed there are more straightforward ways to estimate $\kappa$ but most of them do not use data from real cloud bases for such a broad range of conditions as we did. We do not suggest methodology is better to constrain $\kappa$ and we agree with the referee that our method is even more complicated and 'convoluted' than others. However, this is in fact one of the main points of our paper to demonstrate the uncertainties in measurements and how they translate into predictions of cloud droplet number concentrations. To our knowledge there are not many studies available that use such a rich set of measurements (e.g. two sets of $N_d$, $w$ analysis) and perform a detailed analysis of their importance for $N_d$ prediction at cloud bases.

c. The last sentence of the abstract says that "Our results indicate that Aitken mode particles and their hygroscopicity can be important for droplet formation at low pollution levels and high updraft velocities in tropical convective clouds." This seems like a result worthy of publication, if the authors first acknowledge previous related work: Leaitch, W. R. et al., Effects of 20–100 nm particles on liquid clouds in the clean summertime Arctic, Atmos. Chem. Phys. 16, 11107–11124 (2016); Baccarini et al., Frequent new particle formation over the high Arctic pack ice by enhanced iodine emissions, Nature Comm. https://doi.org/10.1038/s41467-020-18551-0, 2020.

We thank the referee for these additional references. We would like to refer to our recent publication that includes a detailed discussion on conditions under which Aitken mode particles contribute to CCN (Pöhlker et al., 2021). While the referee is right that under Arctic conditions, i.e. when $N_a$, this might be the case, the parameter space is much broader as it depends on combination of $\kappa$, $N_a$ and $w$. Since the conditions in the Arctic are substantially different to those in our current study in terms of aerosol loading, $w$ and CCN-limited regime, we did not add the references to the manuscript.

d. The last sentence of Section 1 tells us that the modelling is used to examine the sensitivities of Nd to kappa, Na and w, but we've been subjected to such sensitivity studies for many years now, dating back to at least 1971 (Junge and McLaren, JAS, 1971), which told us that the number distribution was more important than the chemical content of the particles. Why is another sensitivity study of this important?

The referee is right that the relative importance of aerosol parameters for cloud droplet number concentration as identified in this study is not new, as it is discussed in Section 4.3. Many of the studies cited in this section are based on idealized cases or conditions.

The novelty of our study is the application of the unique combination of measurements. While at a first glance the two $N_d$ measurements disagree substantially, we demonstrate that both measurements can be reconciled by using a new method that take the advantage of new in situ measurements below and within convective cloud bases. Such type of analysis were not performed as it is presented in this manuscript and over the Amazon basin.

2) Lines 42-50: Leaitch et al., Cloud albedo increase from carbonaceous aerosol, Atmos. Chem. Phys., 10, 7669-7684, 2010 is relevant here.

We thank the referee for this additional reference and added it in the new version.

3) Line 93 – What is the Aerosol Measurement System, and why do we need to know its acronym?

This acronym has been used regularly for data from the HALO aircraft. Thus, we also use it here to make our data sources as transparent as possible and so that the reader can make the connection to other data from the HALO database. We reworded the paragraph as follows to make it clearer that this measurements system measured the total aerosol particle number concentration as used in this study (l. 100):

*The total particle number concentration in the size range of ~ 10 nm to ~ 500 nm ($N_{CN}$) below cloud base was measured using the Aerosol Measurement System (AMETYST). The system is composed of scanning mobility particle sizers and butanol-based condensation particle counters (CPCs, modified Grimm CPC 5.410 by Grimm Aerosol Technik, Ainring, Germany) with a flow rate of 0.6 L min$^{-1}$. Particle losses in the sampling lines have been estimated and taken into account with the particle loss calculator by von der Weiden et al. (2009). Typical uncertainties of CPC measurements are on the order of ~10 % (Petzold et al., 2011; Andreae et al., 2018).*

4) Lines 104-106 – Your inferred Aitken distribution has a minimum at about 80 nm, based on the Wex, Gong and Quinn distributions. The below-cloud distributions measured over the North Atlantic Ocean by Leaitch et al. (ACP, 2010), also used in parcel model calculations, had minima at about 100 nm. Such a minimum might make a substantial difference in your results, for example, potentially requiring a smaller kappa for the Aitken mode.

Many thanks for this comment. In our recent publication by Pöhlker et al. (2021) we indeed showed a large sensitivity of the mode diameter of the Aitken mode to the activation of Aitken mode particles. However, we are confident that the minimum between the two modes in the current study was below 92 nm; a minimum at larger sizes would have shown a different shape in the UHSAS measurements. We added the following text (l. 114):

*Other shapes of marine aerosol size distributions, e.g. as reported by Leaitch et al. (2010), were not considered for our lognormal fit because they were not in agreement with the measured UHSAS data.*

5) Lines 118-126 – Are the differences in the sampling approach by the CAS and CCP potentially responsible for the bias in the two measurements?

Yes, but this is not the only reason. As mentioned above the probes were also mounted on different wings, ~20 m apart from each other. We performed a detailed analysis about the measurements of CAS and CDP in our previous study Braga et al. (2017). Our results indicated that the agreement between CAS and CDP is good assuming the probe uncertainty ranges.

6) Lines 131-133 – Given the better agreement with the King probe, why were the CAS measurements not chosen to compare to the model alone? Were number or sizing differences mostly responsible for the differences between the CAS and CCP probe comparisons with the King probe?

We are somewhat confused about this suggestion as it apparently contradicts the referee's introductory comments (*I am not being critical of the Nd measurements: I applaud the authors for including both.*. We do agree with this referee comment that the consideration of both data sets our study apart from others that base their conclusions from $N_d$ closures on only one date set.

The CAS measurements had better agreement with the King probe due to the fact that the hot-wire was mounted at the CAS probe, and thus, measured the same air sample. Furthermore, the CDP is an open-path instrument, while CAS measure drops that pass the sampling area within a pitot tube. Such characteristics lead to differences in particle sampling, size sensitivities etc. Nevertheless, both probes measured realistic values and the fact that the closure analysis have shown similar results for most of flights enhances the relevance of the use of two probes.

Usually, in other studies, measurements of only one of the probes are available. Thus, our study shows (i) that $N_d$ measurements from different probes might result in different values and (ii) these differences do not translate into significantly different conclusions on the relative importance and absolute values of aerosol parameters to reach $N_d$ closure.

7) Section 3.1 – a) It would be helpful to have an example of the time series of Nd and w during a cloud penetration to demonstrate the application of this approach here. By design, the PMM analysis appears to ensure the connection between Nd and w shown in Figures 3 and 4, whereas other methods, such as Peng that you later refer to as weaker (Lines 339-345), do not. Because of the apparent forced dependence by the PMM method, it seems inappropriate to use this approach to make conclusive statements about the dependence of Nd on w.

We have added a new figure showing the time series of $N_d$ and *w* within cloud. The new text and figure are shown as below (l. 166):

*"...Figure 2 shows an example of measured w at cloud bases and estimated w based on the PMM analysis ($w_{PMM}$). The figure shows that $w_{PMM}$ are in well agreement with measurements at cloud bases. Furthermore, for cloud passes in which the values of w are negative realistic w are estimated based on PMM."*

[Figure]

**Figure R2-1.** Time series of droplet size distribution measured by CCP-CDP [top], Number concentration of droplets ($N_d$) [middle], Measured vertical velocities $w$, and estimated $w$ based on PMM method [bottom]. The measurements were performed during flight AC19 above the Atlantic Ocean.

b) The Peng approach, and perhaps others, were developed for stratus and stratocumulus with relatively low updraft speeds. The pathways of air parcels through such clouds, as shown by modelling studies (e.g., at CSU), are much different than for more convective clouds, and therefore the Peng approach may not be appropriate here in any case. For strong convection, the $w$ may significantly increase with height above cloud base. How do you know that your $w$, measured more than 20 metres above base, were indeed representative of the $w$ at cloud base? The Nd may not change much with height (assuming only homogeneous entrainment), but the $w$ can, and based on the LWC shown in Figure 2 sampling was conducted well above cloud base in many cases, which would lead to overestimating $w$ in some cases.

For cloud base height we assumed the level of maximum supersaturation from the vertical profile of simulated supersaturation. Since the value of supersaturation over water within cloud cannot be measured, the height above cloud base mentioned in the MS regards to model simulation results.

The measurements of w and Nd were performed at cloud bases of cumulus humilis and mediocris. These cloud passes were selected based on the videos recorded by the HALO cockpit forward-looking camera. Therefore, we could not ascribe a specific height for measurements of cloud bases but rather state that they had approximately the same altitude during the cloud passes.

Figure 2 shows the simulated and measured values of LWC. The heights of LWC are only shown for the model results. The frequency of measured LWC is shown for cloud bases regardless of the height above cloud bases.

We clarify these points at Section 2.2 as follows:

"...*We refer in the current study to the measurements closest to cloud base as 'cloud base' measurements, even if the actual cloud base might have been slightly below this altitude of measurements (Section 3.2.2 and Figure 2). The cloud base measurements were selected based on the videos recorded by the HALO cockpit forward-looking camera...*"

We have re-written section 3.2.2 to better explain this point (l. 187 ff):

*The cloud base measurements were performed at approximately constant altitude during each research flight and were selected based on the videos recorded by the HALO cockpit forward-looking camera. However, these measurements might represent different levels in relation to the level of maximum supersaturation at cloud base, which depends on the updraft velocity and turbulence in cloud. In order to determine the height at which $N_{d,m}$ and $N_{d,p}$ should be compared, the measured liquid water content (LWC) was compared to the simulated LWC using the aerosol size distribution for the different flights, together with w measured at cloud base and an assumed hygroscopicity of $\kappa = 0.1$.*

*Under adiabatic conditions, $N_{d,p}$ is approximately constant at $\sim 20$ m above the level of the maximum supersaturation $S_{max}$ (Fig. S5). Figure 2 shows the predicted LWC from the simulations as a function of height above $S_{max}$ for the four flights. Overlaid on the model results (colored lines) are the frequencies of measured LWC by the cloud probes near cloud base (white bars). The measured LWC represents the cumulative mass size distribution. For all flights the model predictions in most of cases match the minimum LWC measured at $\sim 20$ m above the $S_{max}$ level. This height level might represent slightly different absolute heights above the surface and the level of saturation estimated by the model (RH = 100%) (Fig. S6). However, since we focus our discussion in the following section on the comparison of $N_{d,m}$ and $N_{d,p}$, we perform our analysis assuming modeled data for a height of 20 m above $S_{max}$.*

8) Line 190 – Unless there is some inhomogeneous mixing.

The referee is correct that inhomogeneous mixing in cloud might lead to a reduction in $N_d$. However, in the text here, we refer to our model predictions. Inhomogenous mixing cannot be taken into account in our adiabatic parcel model.

9) Lines 207-208 - Curious statement: effectively, you are saying that variations in Nd of 20-40% aren't large enough to worry about. Is that the status of the indirect effects?

We do not fully understand the referee's comment. We stated in the text that the two sets of $N_d$ measurements differ by $\sim$20% and that these values can be reproduced by the model by using very similar $\kappa$ values. In our opinion, this is fully in agreement with the initial referee comment and our response (1.d) and with many previous studies that are cited in Section 4.3 that have shown a weak sensitivity of $N_d$ to $\kappa$.
Since we feel that sensitivities and this parameter ranking are sufficiently discussed in the introduction and in Section 4.3, we did not change any text here.

10) Lines 209-211 - This statement is predicated on the correct answer lying between the two measurements, yet there is nothing in the paper to suggest that is correct.

We did not mean to imply that the 'best value' is in-between those of the two measurements. In fact, we had used averaged values in the previous version of the manuscript (but were rightfully criticised by the referees for doing so). Therefore, we performed our closure study for both measurements separately. As we realized that the text might be misinterpreted in this regard, we changed it as follows (l. 221):

*The use of a single cloud probe might lead to a biased estimate based on the data set of each cloud probe separately. The consideration of both cloud probes shows the uncertainty in $N_d$ measurements and therefore the uncertainty range of $\kappa$ and/or $N_a$ values for $N_d$ closure.*

11) Lines 219-220 – "Confirm" is an exaggeration. It could be true, but you haven't demonstrated this, and there are better ways to estimate kappa. This approach is too uncertain.

We agree with the referee that our approach of deriving $\kappa$ may be more uncertain than others, such as CCN (as opposed to $N_d$) closure studies or direct composition measurements. However, given that we derive similar conclusions as such previous studies suggests that our measurements of $w$ and $N_a$ and their correlations as implied by the PMM approach were sufficiently constrained. We reworded the text as follows (l. 229): *Our results  are in agreement with previous studies. The value range is representative of internally mixed aerosol particle populations during the dry season in the Amazon Basin, which are influenced by fresh and aged biomass burning aerosol from Amazon and Africa.*

12) Lines 222-225 – If you sampled very near cloud base, then it would seem difficult for entrainment to increase the Nd. Again, it is important to illustrate your in-cloud selection process. If you sampled too high above cloud base, then you need to question your estimate of w. Also, your statement here is incorrect if the CCP measurements of Nd are closest to reality. Overall, I don't find this paragraph useful.

We agree with the referee that this text distracted from our main messages. Therefore we removed it.

13) Lines 231-235 – When you say the hygroscopicity of particles could change due to dissolution of soluble compounds, do you mean the uptake of highly soluble gases, such as HNO3? If so, a reference is in order: Kulmala et al., maybe JGR, in the 1990s. If you are talking about delays associated with weakly soluble compounds, then Shantz et al.: Effect of organics of low solubility on the growth rate of cloud droplets, J. Geophys. Res., 108, 4168-4177, 2003 is appropriate here and on line 405. Your assumption that reduced solubility might flatten the curves more may or may not be correct, depending on the entire chemical composition of the size distribution. Shantz et al found that delayed growth reduced the Nd relative to a highly soluble compound. I don't see enough support for your statement that "such composition effects can likely be excluded."

We agree with the referees that there are several aspects that would need to be considered for any speculations regarding such composition effects. Therefore, we removed these two sentences and rather frame our conclusions in the context of an 'effective $\kappa$' as it has been done in previous studies and that encompasses several composition effects.

14) Lines 236-237 – Yet, your Aitken mode is highly soluble.

We do not have any information on the solubility of the Aitken mode particles. However, we do show that various combination of $\kappa_{Ait}$ and $\kappa_{acc}$ can lead to similar $N_{d,p}$. The sentence as written does not exclude the possibility that Aitken mode particles are equally or more hygroscopic than accumulation mode particles.

15) Lines 338-247 – Again, I suggest that your tendency of measured Nd vs w is at least in part a result of your approach.

Yes, our approach, i.e. the PMM, implies that there is a correlation between $N_d$ and *w*. We did not make any changes in the manuscript as the method is explained in Section 3.1 and in more detail in previous publications (Haddad and Rosenfeld, 1997; Braga et al., 2017a)

16) Line 274 – Nd to Na?

Thank you. The referee is right that the text should read:

*...it was demonstrated that the sensitivity of $N_d$ to $N_a$ becomes small...*

17) Lines 350-353 – "we conclude that the sensitivity of Nd to Na is much greater than that to w under these conditions which is also reflected by the rather small increase in Nd with w at high updraft velocities". Again, we need to know where with respect to cloud base that the w were measured, in case some of your w are overestimated.

We think the referee misunderstood where the measurements of w and Nd took place. We tried to better explain that the measurements of w were performed within cloud bases selected based on the videos recorded by the HALO cockpit forward-looking camera (as described in the new Section 3.2.2 and in our response to Comment 7). The exact height of measurements above cloud base (Smax) cannot be estimated. The height above cloud base ($S_{max}$ level) of 20 m was used for modeled $N_d$ for the reasons described in Section 3.2.2. This section was modified now to better explain our methodology to constrain modeled and measured $N_d$ at cloud bases.

**References**

Andreae, M. O., Afchine, A., Albrecht, R., Amorim Holanda, B., Artaxo, P., Barbosa, H. M., Borrmann, S., Cecchini, M. A., Costa, A., Dollner, M., Fütterer, D., Järvinen, E., Jurkat, T., Klimach, T., Konemann, T., Knote, C., Krämer, M., Krisna, T., Machado, L. A., Mertes, S., Minikin, A., Pöhlker, C., Pöhlker, M. L., Pöschl, U., Rosenfeld, D., Sauer, D., Schlager, H., Schnaiter, M., Schneider, J., Schulz, C., Spanu, A., Sperling, V. B., Voigt, C., Walser, A., Wang, J., Weinzierl, B., Wendisch, M., and Ziereis, H.: Aerosol characteristics and particle production in the upper troposphere over the Amazon Basin, Atmospheric Chemistry and Physics, https://doi.org/10.5194/acp-18-921-2018, 2018.

Braga, R. C., Rosenfeld, D., Weigel, R., Jurkat, T., Andreae, M. O., Wendisch, M., Pöhlker, M. L., Klimach, T., Pöschl, U., Pöhlker, C., Voigt, C., Mahnke, C., Borrmann, S., Albrecht, R. I., Molleker, S., Vila, D. A., Machado, L. A. T., and Artaxo, P.: Comparing parameterized versus measured microphysical properties of tropical convective cloud bases during the ACRIDICON-CHUVA campaign, Atmos. Chem. Phys., https://doi.org/doi.org/10.5194/acp-2016-872, 2017.

Leaitch, W. R., Lohmann, U., Russell, L. M., Garrett, T., Shantz, N. C., Toom-Sauntry, D., Strapp, J. W., Hayden, K. L., Marshall, J., Wolde, M., Worsnop, D. R., and Jayne, J. T.: Cloud albedo increase from carbonaceous aerosol, Atmospheric Chemistry and Physics, 10, 7669–7684, https://doi.org/10.5194/acp-10-7669-2010, 2010.

Petzold, A., Marsh, R., Johnson, M., Miller, M., Sevcenco, Y., Delhaye, D., Ibrahim, A., Williams, P., Bauer, H., Crayford, A., Bachalo, W. D., and Raper, D.: Evaluation of Methods for Measuring Particulate Matter Emissions from Gas Turbines, Environmental Science & Technology, 45, 3562–3568, https://doi.org/10.1021/es103969v, pMID: 21425830, 2011.

Pöhlker, M. L., Zhang, M., Campos Braga, R., Krüger, O. O., Pöschl, U., and Ervens, B.: Aitken mode particles as CCN in aerosol- and updraft-sensitive regimes of cloud droplet formation, Atmospheric Chemistry and Physics, 21, 11 723–11 740, https://doi.org/10.5194/acp-21-11723-2021, https://acp.copernicus.org/articles/21/11723/2021/, 2021.

Quinn, P. K. and Coffman, D. J.: Local closure during the First Aerosol Characterization Experiment (ACE 1): Aerosol mass concentration and scattering and backscattering coefficients, , 103, 16,575–16,596, https://doi.org/10.1029/97JD03757, 1998.

von der Weiden, S.-L., Drewnick, F., and Borrmann, S.: Particle Loss Calculator – a new software tool for the assessment of the performance of aerosol inlet systems, Atmospheric Measurement Techniques, 2, 479–494, https://doi.org/10.5194/amt-2-479-2009, https://amt.copernicus.org/articles/2/479/2009/, 2009.

---

## Author Response (AR3)

**Author response acp-2021-80**

**Submitted October 2, 2021**

*Author response: We thank the referee for the detailed comments. Below you find our specific responses (in red) to the referee comments (in black). Manuscript text is in purple, new text in purple bold.*

**Anonymous Referee #3**

*1- Authors' response to my original comment 1): "We respectfully disagree that the term 'closure' is not appropriate here. We refer to the definition of 'closure', as described in previous studies, e.g., by Quinn and Coffman (1998): "In a closure experiment, an aerosol property is measured by one or more methods and calculated from a model that is based on other independently measured properties. A comparison of the measured and calculated values can reveal inadequacies in either the measurements or the model." Even the aforementioned study did not focus on cloud droplet number concentration, generally this definition is very well in agreement with our approach."*

*Review comment on this response: - Of course, I agree with the stated reference as to the nature of closure. Unfortunately, I find this statement to reinforce my point, rather than the authors. You have not measured kappa by any methods. You use the model to make an assessment of kappa, but you cannot call it closure because you do not have anything to give you an independent assessment of kappa to compare against.*

Author response: The reviewer is correct that we did not use direct kappa measurements that were performed during the same field campaign. However, as we point out throughout our manuscript, we used kappa values from other studies at the same location and in other comparable air masses (e.g. marine). These kappa values were determined independently from the measurements of the other aerosol properties, namely particle number concentration.

Our model was NOT set up such that we fitted kappa to get the best agreement of measured and modeled cloud droplet number concentration without any constraints on the kappa value. In that case, we would agree with the referee that such a model exercise should not be called 'closure', but 'finding the best fitted value'.

The assumptions made in our model studies were guided therefore by previous measurements as indicated throughout the manuscript, e.g.

l. 11: assuming an average hygroscopicity of $\kappa \sim 0.1$, which is consistent with Amazonian biomass burning and secondary organic aerosol

l. 227 ff: This $\kappa$ value has been suggested previously for comparable air masses during the dry season in the Amazon Basin (e.g., Pöhlker et al., 2016, 2018). In these prior studies, $\kappa$ was constrained based on size-resolved CCN measurements and measurements of the aerosol chemical composition, dominated by an aged organic fraction. [...] The value range is representative of internally mixed aerosol particle populations during the dry season in the Amazon Basin, which are influenced by fresh and aged biomass burning aerosol from Amazon and Africa.

l. 270: in the absence of more information on the particle hygroscopicity we cannot state with certainty that the assumptions of the two values are appropriate for this aerosol population.

*2- Authors' response to my original comment 1 b): "The referee is right that indeed there are more straightforward ways to estimate kappa but most of them do not use data from real cloud bases for such a broad range of conditions as we did."*

*Review comment on this response: Why would there be a difference between estimating kappa below a cloud rather than not below a cloud? Effects on photochemistry might play some role, but I don't see how your results that show a range of kappa values is either better or for that matter distinguishable from an approach that offers a more direct estimate of kappa, whether below or not below a cloud.*

Author response: We are afraid that the referee misunderstood our previous response and the single sentence he cites was taken out of the context. We do not imply that the kappa values below and in cloud are different. (Even though this may be the case due to dissolution of soluble particle components or other effects but this is not topic of our study.)

For completeness, we repeat our complete response to the referee's previous comment here.

*"The focus of the paper is to describe the closure analysis at cloud bases of convective clouds. The referee is right that indeed there are more straightforward ways to estimate κ but most of them do not use data from real cloud bases for such a broad range of conditions as we did. We do not suggest methodology is better to constrain κ and we agree with the referee that our method is even more complicated and 'convoluted' than others. However, this is in fact one of the main points of our paper to demonstrate the uncertainties in measurements and how they translate into predictions of cloud droplet number concentrations. To our knowledge there are not many studies available that use such a rich set of measurements (e.g. two sets of Nd, w analysis) and perform a detailed analysis of their importance for Nd prediction at cloud bases*

Based on this, we would like to re-emphasize the novelty of our study to use **cloud droplet measurements, paired with fairly well constrained updraft measurements.** Most of the past closure studies were performed using CCN measurements in CCN counters, i.e. at equilibrium conditions. Such measurements mimic conditions below cloud, i.e. when particles take up water vapor but have not grown to cloud droplets yet.

Compared to such *CCN closure studies*, the number of *cloud droplet number (Nd) closure studies* is much smaller since cloud properties, such as drop number concentration and co-located updraft velocities, are much more challenging to measure. However, by our analysis, using PMM, we reduced the uncertainty in such Nd closure studies by better constraining w and therefore reducing the uncertainty.

This is stated, e.g. in l. 409ff: *Implying that higher Nd are formed in regions of higher updraft velocities, we sorted observed data of Nd and w by their frequency of occurrence ('probability matching method'). Using this approach, we reduced the uncertainty of w for the Nd closure. Therefore, we could largely limit our sensitivity analysis to the investigation of the importance of particle hygroscopicity and number concentration for cloud droplet number concentrations.*

*3- Authors' response to my original comment labelled 1c: "Since the conditions in the Arctic are substantially different to those in our current study in terms of aerosol loading, w and CCN-limited regime, we did not add the references to the manuscript."*

*Review comment on this response: This is about processes, not location. The relative importance of the many relevant processes may vary from one location to another, but in this case one of the most important factors is the effect of "aerosol loading". Low concentrations of larger particles are the main reason Aitken particles can activate in the Arctic. As for updrafts, if particles as small as 20-30 nm can activate in the Arctic, then it is more likely to happen in cases of higher updrafts. These references support your work here, and they are warranted.*

Author response: The referee is correct that our response was not really clear. Of course, the conclusions regarding a possible contribution of Aitken mode particles to CCN is not depend on location but on the parameters and processes.

In our previous paper (Pöhlker et al., 2021), we discuss in detail that the role of the Aitken mode particles depend on a combination of aerosol loading and updraft velocity. We would like to point out that our study focuses on cumulus clouds whereas the studies the referee cites here are on stratus clouds for which the Na/w thresholds are different above which Aitken mode particles contribute to CCN.

To make this clear, we added in l. 366 (new text in bold)

*Qualitatively this was also suggested in a previous Nd closure study for marine stratocumulus clouds, where it was concluded that only the presence of an Aitken mode could explain the high Nd,m at updraft* **velocities of w ≥ 1 m s-1 (Schulze et al., 2020)). Generally, the conditions at which Aitken mode particles contribute to CCN depend on the combinations of the parameter values of Na, w and κ (Pöhlker et al., 2021). Therefore, Aitken mode particles were shown to contribute to CCN in Arctic stratocumulus clouds or fog, that are characterized by low w and (Jung et al., 2018; Korhonen et al., 2008; Leaitch et al., 2016) whereas both updraft and aerosol loading are much higher in the convective cumulus clouds in the Amazon.**

*4- In response to review comment 7b), the authors state: "we could not ascribe a specific height for measurements of cloud bases but rather state that they had approximately the same altitude during the cloud passes".*

*Review comment on this response: Thank you for Figure R2-1. You should be able to calculate an approximate distance above cloud base based on the size of the droplets using your adiabatic parcel model. LWC is dependent on height above base, not updraft speed, so you just need to define the activated number concentrations and see at what heights you reach your measured diameters. You don't even need a lot of sophistication, and this will offer a better estimate of height above base.*

Author response: We thank the referee for their suggestion. This is exactly the procedure we applied in order to constrain the height in cloud at which we compared the measured and predicted cloud droplet number concentration:

- Using our parcel model, we calculated the LWC based on the predicted droplet number concentration and droplet size.

- The resulting LWC was then compared to the measured LWC

- The predicted height above cloud base using our adiabatic parcel model at which both the predicted and measured LWC showed best agreement was then defined as the 'approximate distance above cloud base' at which our closure was performed.

We did not perform a comparison of predicted and measured cloud droplet sizes but, other than that, it seems that our approach is what the referee was suggesting and which led us to the conclusion that a height of ~20 m above cloud base is an appropriate value. Since we dedicated a full Section (3.2.2 Determination of in-cloud height to compare Nd,m and Nd,p) and Figure 3 to this, we did not add any further description to the text.

*5- In response to review comment 14 ("Yet, your Aitken mode is highly soluble"), the authors state that they have no information on the solubility of Aitken mode particles. On lines 259-261 of the revised paper, they state that "Even assuming rather extreme values of kappaAit = 0.8 cannot fully reproduce the large increase in Nd at w & 1.5 m s-1 as observed by the CAS probes; assuming very hygroscopic Aitken mode and less hygroscopic accumulation mode particles can approximately reproduce the trend in Nd,m from the CDP."*

Author response: We are not fully sure that we understand the referee's comment. Does the referee would like to point out an apparent contradiction?  We do not conclude that the Aitken mode was

indeed highly soluble. It should be kept in mind that we use kappa as an "effective parameter, encompassing all factors that affect water uptake" (l. 177, and in agreement with previous studies). To make this clearer, we added at l. 261 (new text in bold)

*Even assuming rather extreme values of Ait = 0.8 cannot fully reproduce the large increase in Nd at w ≥1.5 m s-1 as observed by the CAS probes; assuming very hygroscopic Aitken mode and less hygroscopic accumulation mode particles can approximately reproduce the trend in Nd,m from the CDP.* **It should be kept in mind that $\kappa$ is considered an effective parameter that may also reflect water uptake due to additional processes or effects that are not represented in our model and therefore cannot be further reconciled here.**

*6- Review comment 15 – The authors acknowledge that the approach they use to associate Nd and updraft speed reinforces a strong correlation. In other words, the approach is not entirely objective. This should at least be discussed somewhere in the paper.*

Author response: The referee is correct that the Probability matching method (PMM) applied here implies a correlation of Nd and w. Strictly, this approach is therefore indeed not entirely objective but implies the assumption that one of the parameters affects the other. We do not think that we need to defend this assumption as it follows from the equation that is generally accepted as a basis for cloud physics (Twomey, 1959).

$N_{CCN} = N_0 \cdot S^k$

whereas the supersaturation S generally increases with updraft velocity (with all else being equal). Deviations from this relationship may be caused, for example, due to entrainment. However, near cloud base this is unlikely. While this had been discussed in the cited references (Braga et al., 2017), it was not clearly pointed out in the current study. Therefore, we added in l. 160 (new text in bold):

*The PMM analysis is based on the assumption that these two related variables increase monotonically with each other.* **This assumption implies that entrainment – which may lead to a reversal of the assumed trend - can be neglected near cloud base which is likely a valid assumption under these conditions.**

*7– I do applaud including both measurements of Nd, but I feel that one is likely to be closer to the truth than the other, and I think this could be assessed.*

Author response: We are not sure what the referee is proposing here. Based on our analysis, we cannot assess whether one Nd measurement is 'closer to the truth' than the other. In fact, we are not even sure how to define 'truth' in this context since the two probes use different characteristics to determine the droplet number (cf. Section 2.2). However, we are convinced that reporting data from both probes is very valuable as it demonstrates that there may be biases in conclusions if only data from one of the probes are considered. Since this was also emphasized by the referee in their last report and repeated here ("*I do applaud including both measurements of Nd*") and also made clear throughout paper, e.g.

*l. 63: This [i.e. the closure analysis] was performed to verify our methodology using two types of instruments to measure number concentrations of droplets with different particle inlet characteristics and uncertainties*

*l. 219: The deviations between Nd,m from CCP-CDP and CAS-DPOL (~ 21% on average) reinforce the advantage of duplicate measurements for the closure analysis. The use of a single cloud probe might lead to a biased $\kappa$ estimate based on the data set of each cloud probe separately.*

References

Jung, C. H., Yoon, Y. J., Kang, H. J., Gim, Y., Lee, B. Y., Ström, J., et al. (2018). The seasonal characteristics of cloud condensation nuclei (CCN) in the arctic lower troposphere. *Tellus B: Chemical and Physical*

*Meteorology*, *70*(1), 1–13. https://doi.org/10.1080/16000889.2018.1513291

Korhonen, H., Carslaw, K. S., Spracklen, D. V, Ridley, D. A., & Ström, J. (2008). A global model study of processes controlling aerosol size distributions in the Arctic spring and summer. *Journal of Geophysical Research: Atmospheres*, *113*(D8). https://doi.org/https://doi.org/10.1029/2007JD009114

Leaitch, W. R., Korolev, A., Aliabadi, A. A., Burkart, J., Willis, M. D., Abbatt, J. P. D., et al. (2016). Effects of 20--100\,nm particles on liquid clouds in the clean summertime Arctic. *Atmospheric Chemistry and Physics*, *16*(17), 11107–11124. https://doi.org/10.5194/acp-16-11107-2016

Pöhlker, M. L., Zhang, M., Campos Braga, R., Krüger, O. O., Pöschl, U., & Ervens, B. (2021). Aitken mode particles as CCN in aerosol- and updraft-sensitive regimes of cloud droplet formation. *Atmospheric Chemistry and Physics Discussions*, *2021*, 1–26. https://doi.org/10.5194/acp-2021-221

Schulze, B. C., Charan, S. M., Kenseth, C. M., Kong, W., Bates, K. H., Williams, W., et al. (2020). Characterization of Aerosol Hygroscopicity Over the Northeast Pacific Ocean: Impacts on Prediction of CCN and Stratocumulus Cloud Droplet Number Concentrations. *Earth and Space Science*. https://doi.org/10.1029/2020EA001098

Twomey, S. (1959). The nuclei of natural cloud formation part II: The supersaturation in natural clouds and the variation of cloud droplet concentration. *Geofisica Pura e Applicata*, *43*(1), 243–249. https://doi.org/10.1007/BF01993560